



# Oxygen and light determine the pathways of nitrate reduction in a highly saline lake

Nicolás Valiente[1,2], Franz Jirsa[3,4], Thomas Hein[5,6], Wolfgang Wanek[7], Patricia Bonin[8], Juan José Gómez-Alday[2]

[1]Centre for Biogeochemistry in the Anthropocene, Department of Biosciences, Section for Aquatic Biology and Toxicology, University of Oslo, PO Box 1066 Blindern, 0316 Oslo, Norway
[2]Biotechnology and Natural Resources Section, Institute for Regional Development (IDR), University of Castilla–La Mancha (UCLM), Campus Universitario s/n, 02071 Albacete, Spain
[3]Institute of Inorganic Chemistry, University of Vienna, Waehringer Strasse 42, 1090 Vienna, Austria
[4]Department of Zoology, University of Johannesburg, PO Box 524, Auckland Park, 2006 Johannesburg, South Africa
[5]WasserCluster Lunz – Inter-university Center for Aquatic Ecosystem Research, Lunz am See, Dr. Carl Kupelwieser Prom. 5, 3293 Lunz/See, Austria
[6]Institute of Hydrobiology and Aquatic Ecosystem Management, Department of Water, Atmosphere and Environment, University of Natural Resources and Life Sciences, Gregor-Mendel-Str. 33, 1180 Vienna, Austria
[7]Division of Terrestrial Ecosystem Research, Department of Microbiology and Ecosystem Science, University of Vienna, Althanstrasse 14, 1090 Vienna, Austria
[8]Aix-Marseille Université, CNRS, Université de Toulon, IRD, MIO UMR 110, 13288 Marseille, France

*Correspondence to*: Nicolas Valiente (n.v.parra@ibv.uio.no)

**Abstract.** Nitrate ($NO_3^-$) removal from aquatic ecosystems involves several microbially mediated processes including denitrification, dissimilatory nitrate reduction to ammonium (DNRA), and anaerobic ammonium oxidation (anammox) regulated by slight changes in environmental gradients. Saline lakes are prone to the accumulation of anthropogenic contaminants, making them highly vulnerable environments to $NO_3^-$ pollution. We investigated nitrate removal pathways in mesocosm experiments using lacustrine, undisturbed, organic-rich sediments from Pétrola Lake (Spain), a highly saline waterbody subject to anthropogenic $NO_3^-$ pollution. We used the revised $^{15}N$-isotope pairing technique ($^{15}N$-IPT) to determine $NO_3^-$ sink processes. Our results demonstrate the coexistence of denitrification, DNRA, and anammox processes, and their contribution was determined by environmental conditions (oxygen and light). DNRA and $N_2O$-denitrification were the dominant nitrogen (N) removal pathways when oxygen and/or light were present (up to 82%). In contrast, anoxia and darkness promoted $NO_3^-$ reduction by DNRA (52%) and N loss by anammox (28%). Our results highlight the role of coupled DNRA-anammox, as yet has never been investigated in hypersaline lake ecosystems. We conclude that anoxia and darkness favored DNRA and anammox processes over denitrification and therefore reduce $N_2O$ emissions to the atmosphere.

## 1 Introduction

Nitrogen (N) is an essential component of living organisms and its availability controls the function of aquatic ecosystems. Since the invention of technical N-fixation through the Haber–Bosch process, humankind has drastically modified the global



N budget, significantly increasing the global fixed N pool and nitrous oxide ($N_2O$) emissions to the atmosphere (Canfield et

al., 2010). Among N species, nitrate ($NO_3^-$) is a widespread compound responsible for water degradation due to excessive fertilizer use in agriculture (Spalding and Exner, 1993). $NO_3^-$ accumulation can increase primary production in surface waters and, as a consequence, can trigger oxygen deficiency and promote eutrophication of surface waterbodies (Vitousek et al., 1997). Among aquatic ecosystems, saline lakes are highly vulnerable to $NO_3^-$ pollution. These ecosystems are mainly located in closed hydrological systems from arid and semi-arid regions, which combined with low precipitation and high

evaporation rates typical of arid climates, lead to accumulate and biomagnify many pollutants compared to freshwater systems (Williams, 2002).

Microbial processes are controlling the Earth's N cycle for ~2.7 billion years and have been widely studied in aquatic ecosystems. Certain microorganisms (diazotrophs) are able to fix $N_2$ into a biologically useful form. In shallow aquatic ecosystems, cyanobacteria often dominate benthic $N_2$ fixation, which is tightly linked with light availability and

photosynthesis (Lu et al., 2018). Moreover, the role of cyanobacterial blooms altering N attenuation within aquatic ecosystems has been recently reported (Zilius et al., 2018). In contrast to diazotrophs, microbes performing denitrification and anaerobic ammonium oxidation (anammox) are responsible for removing fixed N by producing $N_2$ gas (Canfield et al., 2010).

Denitrification is considered to be the primary process of $NO_3^-$ removal from aquatic environments whose main end product

is dinitrogen gas (hereafter referred to as $N_2$-denitrification) through a multi-step reduction process (Harrison et al., 2009; Fernandes et al., 2016; Kuypers et al., 2018). $N_2O$ is an obligate intermediate of denitrification which can be also its main end product (hereafter referred to as $N_2O$-denitrification) and $N_2O$ comprises an important atmospheric greenhouse gas (310 times more potent than carbon dioxide) (Trogler, 1999). However, the nitrification process ($NH_4^+$ oxidation to $NO_3^-$) is also a significant source of atmospheric $N_2O$ as a by-product (Bremner and Blackmer, 1978; Dore et al., 1998; Löscher et al.,

2012). A large diversity and potential activity of denitrifying bacteria have been previously shown for saline lakes (Kulp et al., 2007; Lipsewers et al., 2016), as well as the existence of denitrification at the field scale (Doi et al., 2004; Gómez-Alday et al., 2014). In such ecosystems, variable redox conditions and the supply of organic matter (OM) and nutrients lead to increased $N_2O$ production by denitrification (Huttunen et al., 2003; Liu et al., 2015). In fact, $N_2O$ reduction to $N_2$ seems to be a rate-limiting step during denitrification at extremely high salinities (Shapovalova et al., 2008). Denitrification is strongly

affected by oxygen availability. Despite nitrous oxide reductase activity has been always considered to be inhibited already at relatively low oxygen concentrations (0.25 mg/L) (Bonin and Gilewicz, 1991), recent studies showed that Clade II $N_2O$ reductases (NosZ), abundant in many biomes, are highly expressed and able to reduce $N_2O$ in the presence of low $O_2$ concentrations (Yoon et al., 2016; Hallin et al., 2018).

Under anaerobic conditions, anammox couples ammonium ($NH_4^+$) oxidation to nitrite ($NO_2^-$) reduction to produce $N_2$ (Van

de Graaf et al., 1995). The activity of anammox bacteria has been described in marine ecosystems (Thamdrup and Dalsgaard, 2002), including deep-sea hypersaline anoxic basins (Van der Wielen et al., 2005), and inland waters (Schubert et al., 2006; Abed et al., 2015; Roland et al., 2017). To date, only a limited number of studies have identified anammox bacteria in saline



systems (Yang et al., 2012; Lipsewers et al., 2016), with a totally different community structure than described in freshwater lakes (Wang et al., 2015). So far, however, very little attention has been paid to the role of anammox processes in saline

lakes.

When oxygen becomes limiting or unavailable, specific microorganisms can perform dissimilatory nitrate reduction to ammonium (DNRA) to obtain energy, retaining a more assimilable N source in the system. Thereby, by supplying $NH_4^+$ DNRA can promote anammox (coupled DNRA-anammox) (Jensen et al., 2011). In coastal marine environments, DNRA may be as important as, or even more important than, denitrification for $NO_3^-$ reduction (Giblin et al., 2013), even causing

eutrophication of coastal ecosystems during warm periods (Bernard et al., 2015). DNRA is promoted when $NO_3^-$ availability is low or limited regarding available electron donors (Tiedje, 1988; Van den Berg et al., 2015). Therefore, light can impact on coupled DNRA-anammox as light will enhance primary production and the production of dissolved oxygen, which can strongly reduce the coupled DNRA-anammox process. However, the role of coupled DNRA-anammox in saline lakes has not yet been investigated.

In addition, sediment core incubations have been frequently used to quantify denitrification, DNRA, and anammox rates by applying the $^{15}N$ isotope pairing technique ($^{15}N$-IPT) (Risgaard-Petersen et al., 2003; Roland et al., 2017). The $^{15}N$-IPT was firstly applied on sediment cores to quantify $N_2$ production derived from denitrification (Nielsen, 1992). Since then, many studies have focused on discriminating the relative contribution of N processes using $^{15}N$-IPT, including DNRA, and recently coupled DNRA-anammox (Risgaard-Petersen et al., 2003; Holtappels et al., 2011; Hsu and Kao, 2013; Robertson et

al., 2019). Hence, processes such as anammox have been traditionally underestimated. Recently, a set of equations for $^{15}N$-IPT have been provided allowing to estimate the contribution of nitrous oxide production by $N_2O$-denitrification and the contribution of DNRA to $NO_3^-$ reduction (Song et al., 2016; Salk et al., 2017). Prior to this revised methodology, coupled DNRA–anammox was indistinguishable from denitrification based on isotope tracer experiments (Francis et al., 2007). Henceforth, these process estimates and the new IPT approaches are necessary for a complete N balance estimation.

Here, we tested the hypothesis that oxygen and light conditions in the water column can alter the balance between nitrate removal pathways (denitrification, DNRA, and anammox) during sediment incubations. For this purpose, we incubated lacustrine sediments from a eutrophic lake (Pétrola Lake, Spain) and applied the revised $^{15}N$-IPT to confirm and quantify N-cycling rates.

## 2 Methods

### 95 2.1 Study site

Samples were collected from Pétrola Lake (38º 50' 14'' N, 1º 33' 40'' W), 35 km southwest of Albacete, Spain. Pétrola Lake is the main wetland in the endorheic Pétrola–Corral-Rubio–La Higuera Saline Complex, which covers a total area of about 275 km$^2$ and is located in the southeastern Castilla-La Mancha Region, in the High Segura river basin. Pétrola Lake is a terminal lake occupying about 1.76 km$^2$ in the lowest topographic position of the Pétrola endorheic basin (43 km$^2$), which

consists mainly of Mesozoic materials (Valiente et al., 2017). The endorheic basin is located in a zone vulnerable to eutrophication, and fertilizer use is restricted (Order 2011/7/2 CMA). The lake is shallow (maximum depth 2 m) with major water volume oscillations depending on seasonal precipitation. The hydrofacies varies between Mg-Cl-SO$_4$ (early spring) and Mg-Na-Cl-SO$_4$ (early fall). The piezometric level of the aquifer is close to the topographic surface. Accordingly, various small springs and streams drain the aquifer and discharge into the lake following a radial pattern. The lake has been

classified as a heavily modified water body due to the inputs of agricultural pollutants as well as untreated wastewater directly spilled from Pétrola Village. The Pétrola endorheic basin was declared vulnerable to NO$_3^-$ pollution by the Regional Government of Castilla-La Mancha in 1998. Excess of N in the lake-aquifer system is mainly derived from inorganic synthetic fertilizers (Valiente et al., 2018).

The field survey was conducted in July 2015. The sampling site (control point 2651; Valiente et al., 2018) was located close

to the lake's depocenter and is not affected by direct inputs of polluted freshwater streams or wastewaters. To evaluate initial natural conditions (NC), water samples were collected and stored at 4 °C in darkness prior to further analyses. Furthermore, sediment cores (n=3) were taken from the upper 20 cm lacustrine sediment using acrylic coring tubes (5 cm inner diameter, 20 cm length). Coring tubes were capped at top and bottom with silicone rubber stoppers, cooled, and transported to the laboratory. Once there, the top 5 cm of each core was sliced and used for inorganic N-species extraction. Afterwards, these

slices were then frozen at -20 ºC for further analysis.

Mesocosm preparation for core incubations was adapted from previous works (Welti et al., 2012). For this purpose, acrylic mesocosms (40 cm in length, 20 cm in diameter, containing a total volume of 12.6 L) were used for sampling and incubation to guarantee minimal disturbance of the sediment during sampling (n=9). The mesocosm tubes were acid-prewashed and then drilled into the sediment down to approximately 20 cm depth. Then, mesocosms were filled with 2 L of lake water to

maintain sediment saturation during transport. Additional lake water was collected from the sampling point and stored at 4 ºC to fill the mesocosms in the lab. Black plastic sheets were used to cover the mesocosms to prevent light penetration during transport.

## 2.2 Sediment incubations

In the lab, the remaining volume of each mesocosm was filled with lake water (approximately 4 L), avoiding an air space

over the water column. All mesocosms were sealed gas-tight. The bottom of each mesocosm was firmly sealed with a plate bolted to the mesocosm tube. The top screw cap was equipped with two holes: one to inject air or argon, and the other to collect samples from the water column. To maintain oxic conditions in the water column, air was bubbled using an air compressor connected to a Teflon tube (7 mm inner diameter). From this tube, branches were connected to each mesocosm using a rubber cap. The end of each tube was assembled with a glass Pasteur pipette and placed in the upper water column.

For argon bubbling, a 25 L gas cylinder (Carburos Metalicos, Spain) with a pressure regulator was used. The argon connection to each mesocosm was similar to that used for air bubbling.





For sample collection, a Teflon tube (4 mm inner diameter) was installed through the hole using a rubber cap. The tube inlet was placed 1 cm over the sediment surface, whereas the tube outlet was closed to the atmosphere with a three-way valve. In order to maintain water circulation inside each mesocosm, a small pump was installed in the inner wall. Mesocosms were

placed in a temperature-controlled room (25 ºC) with no exposure to direct sunlight.

Three different treatments were studied in triplicate. Treatment 1 (OL; oxygen + light) mimicked field conditions by means of atmospheric air bubbling, to provide oxygen, and normal dark-light cycles (~ 14 h of light per day; no additional light source was used). Mesocosms of treatment 1 (n=3) were placed close to the room window. OL is henceforth considered as control. For treatment 2 (OD; oxygen + darkness), oxic conditions in the water column were preserved via atmospheric air

bubbling. However, each mesocosm was covered with aluminum foil to protect it from light. Finally, treatment 3 (AD; anoxia + darkness) maintained anoxic conditions by bubbling argon and mesocosms were shielded from light. The bubbling fluxes applied in the experiments were established based on the maximum solubility values of $N_2$ (Hamme and Emerson, 2004) and $N_2O$ (Weiss and Price, 1980) in seawater using a salinity value of 50 g/L, similar to the one previously reported in Pétrola Lake (Valiente et al., 2018). Mesocosms were stabilized in the laboratory until constant $N-NO_3^-$ and $N-NO_2^-$

concentrations in the water column were reached. During the stabilization period, physico-chemical parameters, and inorganic N-species were monitored at 12 h intervals, starting 12 h after collection of the sediment cores (time -36), and finishing 48 h after field sampling (time 0).

In order to apply the $^{15}N$-IPT approach to quantify $NO_3^-$ transformation processes inside the mesocosms, $^{15}N$-labeled nitrate ($K^{15}NO_3$) was added once mesocosm stabilization was reached (time 0). This involved spiking with 250 µmol of $^{15}N-NO_3$,

reaching a water column concentration of about 40 µM $N-NO_3^-$. After labeled nitrate addition, the sampling frequency and incubation times were calculated following the NICE handbook (Dalsgaard et al., 2000). Thus, 30 min intervals were adopted as the initial sampling rate: this was calculated as the optimal time to enable denitrification to reach 90% of its steady state value, assuming a sediment penetration depth of oxygen of 1 mm based on previous works (Valiente et al., 2017).

In each mesocosm, water samples were taken from the water column for inorganic N-species and N-isotope analysis ($N-NO_3^-$, $N-NH_4^+$, $N_2$, and $N_2O$) at times 0, 0.5, 1, 1.5, 2, 2.5, 3, 4, 5, 6, 8, 10, 12, 15, 18, 24, 30, 36, 48, 60, and 72 h with a 50 mL syringe. Moreover, water samples for physico-chemical analyses, dissolved organic carbon (DOC), and dissolved bound nitrogen (DNb) determination were collected at times 0.5, 2, 4, 8, 12, 24, 48, and 72 h from each collecting Teflon tube using a 50 mL syringe. At the end of the incubations, sediment samples were obtained from the upper 5 cm of each mesocosm,

homogenized using a spatula, and used fresh for chemical analyses. Sediment samples were frozen (-20 ºC) before further analyses.

## 2.3 Physico-chemical analyses

Physico-chemical parameters measured included temperature, pH, electrical conductivity (EC), total dissolved solids (TDS), redox potential (Eh), and dissolved oxygen (DO). These parameters were determined directly in the surface water from site



2651 using a HQ40d Portable Multi-Parameter Meter (Hach Company, USA). During sediment incubations, physico-chemical parameters were measured in the collected water samples. Collected water samples were immediately filtered through a 0.45 μm nylon Millipore® filter. Inorganic N-species were determined directly after collection at the Institute for Regional Development (University of Castilla-La Mancha). Determination of $NO_2^-$ and $NO_3^-$ concentration was achieved by UV-VIS spectrophotometry via the modified Griess reaction assay as described by García-Robledo et al. (2014). $NH_4^+$

concentrations were quantified by UV-VIS spectrophotometry using the modified indophenol method, as described by Hood-Nowotny et al. (2010). Dissolved inorganic nitrogen (DIN) was calculated by summing up the concentrations of N-$NO_2^-$, N-$NO_3^-$, and N-$NH_4^+$. DOC and DNb measurements were performed using a Shimadzu TOC-V Analyzer with a total nitrogen measurement unit (TNM-1) at the Institute of Inorganic Chemistry of the University of Vienna, Austria. Dissolved organic nitrogen (DON) concentrations were estimated by subtracting DIN from the measured DNb.

Sedimentary N-$NO_3^-$ (S-N-$NO_3^-$), N-$NH_4^+$ (S-N-$NH_4^+$), and N-$NO_2^-$ (S-N-$NO_2^-$) were determined after extraction from fresh sediment following Hood-Nowotny et al. (2010). Frozen sediment samples were lyophilized for 48 h, followed by homogenization in a porcelain mortar and sieving through a 1 mm steel sieve. Organic matter (OM) content in dried sediment samples was determined as loss of ignition (LOI) by combustion of dried sediments for 2 h at 550 °C at the Institute of Inorganic Chemistry of the University of Vienna, as described by Nelson and Sommers (1996).

**2.4 Isotope composition of N species**


The isotopic composition of N-$NH_4^+$ in the water column was determined by a microdiffusion method (Lachouani et al., 2010). Sample aliquots (10 mL) were transferred to 20 mL HDPE vials with pre-weighed MgO (100 mg). Then, acid traps were added. They trapped the ammonia gas produced from ammonium when hydration of MgO increased the pH to >9.5. Acid traps consisted of glass fiber filter discs (5 mm in diameter) placed on a strip of Teflon tape; 5 μL 2.5 M $KHSO_4$ was

pipetted onto the filter discs and the Teflon tapes folded and closed. Microdiffusion vials were then closed and placed on an orbital shaker at room temperature for 2 days. Subsequently, each acid trap was transferred into a 1.5 mL reaction tube. Tubes were placed into a desiccator containing concentrated $H_2SO_4$ for at least 24 h until further processing. To measure the isotopic composition of N-$NO_3^-$, nitrate was isolated from the previously microdiffused extracts by a reduction-microdiffusion method after conversion by Devarda's alloy to N-$NH_4^+$ (Prommer et al., 2014). The recovery efficiency of the

conversion was expected to be ≥ 95% (Sørensen and Jensen, 1991; Mulvaney et al., 1997). Immediately after adding Devarda's alloy, a new acid trap was added to each vial to retain ammonia gas deriving from nitrate reduction; the subsequent procedure was the same as described above. The filter discs from the acid traps were finally transferred into tin capsules and directly analyzed for N content and at %[15]N by EA-IRMS using an elemental analyzer (EA 1110, CE Instruments) connected via a ConFlo III interface (Thermo Fisher) to a DELTA[plus] IRMS (Finnigan MAT) in the SILVER

Lab (University of Vienna).

To measure the isotopic composition of $N_2$ and $N_2O$, water samples were collected by 60-mL plastic syringes and transferred to gas tight vials (22 mL Exetainer Labco, High Wycombe, UK) containing 1 mL 100 mM $HgCl_2$ to halt biological





reactions. Each vial was completely filled with water sample avoiding any gas headspace. All vials were stored and shipped to the Mediterranean Institute of Oceanography (Aix-Marseille Université) for the analysis of $N_2$ ($^{29}N_2$ and $^{30}N_2$) and $N_2O$

isotopic species concentrations ($^{44}N_2O$, $^{45}N_2O$, and $^{46}N_2O$) using GC-MS (Stevens et al., 1993). Dissolved $N_2$ and $N_2O$ were extracted from the samples in the Exetainer vials by introducing a 6 mL He headspace while simultaneously removing 6 mL of sample water. Sample injection was performed using a modified head-space autosampler (TriPlus 300, Thermo Fisher) that involves gas-equilibration at 65 °C for 6 min whilst shaking vigorously, so that more than 98% of the $N_2$ and $N_2O$ equilibrium concentration was attained (Weiss, 1970). GC-MS analysis was performed using an Interscience Compact GC

system equipped with AS9-HC and AG9-HCT columns. The GC conditions were as follows: injector temperature, 140 °C; oven temperature, 60 °C; carrier gas flow rate, He 15 mL/min; interface temperature, 60 °C. The mass spectrometer was used in electron ionization mode, with an electron energy of 70 eV. Data were acquired in full-scan (m/z 2–200) and selected ion monitoring (SIM) mode (m/z 29, 30 monitored for $N_2$; m/z 44, 45, 46 monitored for $N_2O$). Ar (m/z = 40) was used as an internal standard. Data were acquired and analyzed using Excalibur software. Quantification of N isotopes in both gases was

performed at the Mediterranean Institute of Oceanography (Aix-Marseille Université). Finally, isotopic mass balance calculations were performed using discrete time points compared to the originally added amount of $^{15}N$-$NO_3$. Starting from the initial amount spiked (250 µmol $K^{15}NO_3$), N concentrations and atom percent enrichments were used to calculate the percentage of $^{15}N$ recovery in specific N forms and overall.

**2.5 Denitrification, DNRA, and anammox activity measurements**

For $^{15}N$-IPT modeling, the revised $^{15}N$-IPT calculation procedure (Salk et al., 2017) was applied. A detailed description of parameters and equations is included in Table 1. For this purpose, our incubations were assumed to be intact core incubations. The probabilities of $NO_3^-$ reduction via denitrification, DNRA, and anammox were assumed to be equal (Song et al., 2016). Genuine $N_2$ production via denitrification ($D_{14}$) and anammox ($A_{14}$), as well as $N_2O$ production via denitrification, were calculated for each time step. Production rates were calculated according to Salk et al. (2017) for each

time point after the addition of the labelled $^{15}NO_3^-$. Non-linear increments in the $^{15}N$ content were taken into account by calculating the N production rates (i.e. $^{15}NH_4^+$, $^{29}N_2$, $^{30}N_2$, $^{45}N_2O$, $^{46}N_2O$) from the slope of the initial time point and each specific time point rather than a slope of all time points. Thus, a total of 20 rates of each process were calculated for each mesocosm. Ratios of $^{14}NO_3^-$:$^{15}NO_3^-$ ($r_{14}$) and $^{14}NH_4^+$:$^{15}NH_4^+$ ($r_{14a}$) were calculated and used as base parameters for activity calculations. The applied methodology allowed distinguishing between $N_2$ production via coupled DNRA-anammox and via

canonical anammox. DNRA rates were calculated using the production of $^{15}NH_4^+$, and of $^{30}N_2$ for anammox, over time. However, this model cannot discriminate between $^{15}NO_3^-$ assimilation and subsequent remineralization of OM to $^{15}NH_4^+$, and DNRA. Thus, the DNRA rate may include both processes. The sum of $N_2$ production by denitrification and anammox, together with $N_2O$ production via denitrification, is designated as 'Total N loss'. The 'Total $NO_3^-$ reduction' adds the DNRA rate to the previous estimate.



## 2.6 Statistical analysis

Changes in chemistry and rates of N-loss processes over time as well as at the end of the incubation were assessed using one-way analysis of variance (ANOVA), followed by the Tukey's post hoc test (homogeneous variances) or by the Games-Howell post hoc test (heterogeneous variances). To assess differences in the hydrochemical conditions between initial (n=1) and final conditions (n=9), one-sample two-tailed t-tests were used. Results of statistical tests were considered to be significant at the confidence level 95% ($\alpha = 0.05$). All tests were performed using SPSS-IBM Statistics software.

## 3 Results and discussion

### 3.1 Differences between treatments in chemical parameters and rates of N-loss processes

Differences between initial ($NC_{-48}$, time -48 h) and final conditions ($EX_{72}$, time 72 h) were assessed for the three treatment groups (Table 2). Salinity, as TDS values, was around the hypersaline limit (50 g/L), with values ranging from 45.1 g/L ($NC_{-48}$) to 50.1 g/L ($AD_{72}$). Concerning inorganic N-species in the water column, the final $N-NO_3^-$ and $N-NO_2^-$ concentrations were below the limit of detection (LOD, <0.05 µM). $N-NH_4^+$ concentrations increased significantly (*t*-test, p < 0.05) between $NC_{-48}$ and final conditions in $OL_{72}$ ($t_{(2)} = 8.33$), $OD_{72}$ ($t_{(2)} = 17.89$) and $AD_{72}$ ($t_{(2)} = 19.23$). Furthermore, there was a significant effect of light on the $N-NH_4^+$ concentration ($F_{(2,6)} = 15.98$). Tukey's post hoc tests indicated that the final $N-NH_4^+$ concentration in $OL_{72}$ (139 ± 15.7 µmol/L) was significantly lower than in $OD_{72}$ (175 ± 10.9 µmol/L) and $AD_{72}$ (198 ± 12.2 µmol/L). $N_2$ and $N_2O$ final concentrations (time 72 h) did not show significant differences between treatments ($F_{(2,6)}$ of 0.55 and 0.54, respectively).

DOC concentrations increased significantly between $NC_{-48}$ and final conditions in $OL_{72}$ ($t_{(2)} = 6.30$) and $OD_{72}$ ($t_{(2)} = 9.89$) but not in $AD_{72}$ ($t_{(2)} = 3.79$). Between treatments, there were no significant differences in DOC ($F_{(2,6)} = 0.91$). DNb and DON concentrations did not change over time (p > 0.05), and did not differ between treatments ($F_{(2,6)}$ of 1.28 and 0.95, respectively). The contribution of DON to DNb (DON:DNb) decreased significantly between $NC_{-48}$ and final conditions in all treatments ($OL_{72}$, $t_{(2)} = -26.4$; $OD_{72}$, $t_{(2)} = -6.89$; $AD_{72}$, $t_{(2)} = -8.28$), and differed between treatments ($F_{(2,6)} = 5.31$). Values of pH ($OL_{72}$, $t_{(2)} = -17.14$; $OD_{72}$, $t_{(2)} = -10.26$; $AD_{72}$, $t_{(2)} = -6.43$) and Eh ($OL_{72}$, $t_{(2)} = -7.81$; $OD_{72}$, $t_{(2)} = -8.88$; $AD_{72}$, $t_{(2)} = -515$) decreased significantly between $NC_{-48}$ and final conditions in the three treatments. Between treatments, only pH showed significant differences ($F_{(2,6)} = 5.37$). In the sediment samples, LOI ($F_{(3,8)} = 0.50$) and $S-N-NH_4^+$ ($F_{(3,8)} = 3.54$) did not differ (p > 0.05) between $NC_{-48}$ and final conditions or between treatments. Significant differences were found in $S-N-NO_3^-$ concentrations over time ($F_{(3,8)} = 7.81$), but not between treatments.

Regarding N-loss processes, mean (± standard deviation) and maximum rates are presented in Table 3, whereas a complete record of rates can be found in the Supplementary Information (Table S1). Among treatments, significant differences were only found for DNRA ($F_{(2,161)} = 10.0$). Games-Howell post hoc tests indicated DNRA depends on oxygen in the water



column, distinguishing between AD ($2.80 \pm 2.56$ mmol N m$^{-2}$ h$^{-1}$) and OL ($1.54 \pm 1.53$ mmol N m$^{-2}$ h$^{-1}$) or OD ($1.35 \pm 1.20$ mmol N m$^{-2}$ h$^{-1}$) treatments.

Within each treatment, significant differences were found over time among processes. DNRA and N$_2$O-denitrification showed significant time-related differences in the OL treatment ($F_{(5,47)}$ of 5.70 and 3.82, respectively). These processes, together with N$_2$-anammox in the interval 3-6 h of incubation, were shown as the dominant ones according to Game-Howell

post hoc tests. In the OD treatment, significant differences among processes were found in the interval 3-24 h of incubation. At that time, DNRA and N$_2$O-denitrification rates differed from the other processes ($F_{(5,49)}$ of 6.89 and 3.53, respectively). Games-Howell post hoc tests showed that DNRA was the dominant process in the OD treatment between 3 and 6 h of incubation, and then, up to 24 h of incubation, DNRA was co-dominant with N$_2$O-denitrification. Finally, significant differences were found in the AD treatment between DNRA and the other processes from 3 h of incubation onwards ($F_{(5,50)} =$

3.32). Games-Howell post hoc tests indicated that DNRA was the dominant process up to 48 h.

### 3.2 Hydrogeochemical dynamics during sediment incubations

Three different treatments were applied during sediment incubations by modifying oxygen and light conditions in the water column (see Table 2 for details). The evolution of N parameters is shown in Figure 1 (N-species) and Figure 2 ($^{15}$N). In Figure 1, the evolution of N-NO$_3^-$ and N-NO$_2^-$ showed a well-defined nitrate-reduction pattern in all treatments. N-NO$_3^-$

concentrations decreased to below the limit of detection during the first stage (S1) in all treatments. Immediately after tracer addition, N-NO$_3^-$ increased markedly and then gradually decreased (stage S2), the decline being fastest in the AD treatment (anoxia and darkness). This trend was also observed in Figure 2: $^{15}$NO$_3^-$ reached maximum concentrations at 6 h (OL treatment), 12 h (OD treatment), and 1 h (AD treatment), and was completely removed from the water column within the first 36 h (OL and OD) and 12 h (AD). This decrease in N-NO$_3^-$ concentrations suggests the existence of assimilatory and/or

dissimilatory nitrate reduction processes.

During the final stage (S3), N-NO$_3^-$ was below LOD, which may be the result of either the absence of complete nitrification in the water column, or the rate of NO$_3^-$ consumption being higher than the rate of NO$_3^-$ production. N-NO$_3^-$ in the sediment was significantly higher at the end of the incubations in all treatments. The existence of NO$_3^-$ reduction pathways, mainly during S2, is the most plausible explanation according to the isotopic data. However, other major issues for $^{15}$N-IPT studies

such as uptake and intracellular storage should not be discarded (Robertson et al., 2019). Significant inputs of NO$_3^-$ may also promote blooms of diatoms (frequent in Pétrola Lake), which are physiologically adapted to grow rapidly under nitrate-rich conditions (Bronk et al., 2007). A phytoplankton bloom was observed after $^{15}$N-NO$_3^-$ addition in the light treatment (OL), with a subsequent decay. Even though we cannot prove it, the role of diatom-bacteria aggregates in removing NO$_3^-$ from the water column subsequently fueling benthic anaerobic N-cycling, should be considered (Kamp et al., 2016).

N-NO$_2^-$ peaked during the second stage (S2), paralleling the decrease in N-NO$_3^-$ (Figure 1). Subsequently, N-NO$_2^-$ decreased faster in treatment AD than in treatments OL (oxygen and light) and OD (oxygen and darkness). The intermittent conversion of nitrate to nitrite suggests active NO$_3^-$ reduction processes. In contrast, N-NH$_4^+$ in the water column increased





over time in all treatments. The concentration moderately increased during S1 in all treatments (Figure 1). From the addition

of the labelled $NO_3^-$, concentration of $N\text{-}NH_4^+$ increased (with small oscillations) coupled with a constant increase in $^{15}NH_4^+$

(Figure 2) up to 18 h of incubation. This increase was more pronounced in AD than in OD and OL treatments. From 24 h

until the end of the incubation, $N\text{-}NH_4^+$ concentration increased whereas $^{15}NH_4^+$ tended to stabilize. According to these

results, $N\text{-}NH_4^+$ accumulation in the water column can be explained by DNRA (0 – 18 h), OM remineralization, and

sedimentary $NH_4^+$ release (Kalvelage et al., 2013). During the S1 stage, the absence of $NO_3^-$ hindered the activity of DNRA,

and the increase of $NH_4^+$ in the water column must therefore be a consequence of rapid release from decaying cyanobacteria,

as demonstrated by others (Gao et al., 2013). The small oscillations observed through S2 were the result of fluctuations in $N\text{-}NH_4^+$ production (DNRA, water-column OM remineralization) and consumption (anammox, $NH_4^+$ assimilation, and

nitrification). Thus, $NH_4^+$ accumulation in the water column during S3 can be attributed to sedimentary OM remineralization

after bloom collapse (García-Robledo et al., 2011), as the values did not significantly differ from initial conditions.

Concentrations of $N_2$ were measured from the addition of the $^{15}N\text{-}NO_3^-$. In general, Figure 1 showed a stable concentration

of $N_2$ over time with small peaks in the first 12 h of incubation (positive for OL and AD treatments, negative for OD

treatment). However, a sudden drop in $N_2$ concentration was found at 5 h in treatment OD, probably due to an occasional

episode of $N_2$ stripping, even though steady state gas concentrations were kept throughout the incubations. By comparing

these data with $^{15}N\text{-}N_2$ evolution data (Figure 2) small variations in both $^{29}N_2$ and $^{30}N_2$ were observed after the tracer

addition, where the sharp increase of $N_2$ in the OD treatment at 48 h coincided with an abrupt rise in $^{30}N_2$. During the hours

after the tracer addition (stage S2), the production of $^{30}N_2$ can be attributed either to denitrification or to coupled DNRA-

anammox, by combining the DNRA substrate ($^{15}NO_2^-$) with the DNRA product ($^{15}NH_4^+$) (Holtappels et al., 2011).

Considering $N_2O$ evolution, a different pattern was observed than that described for $N_2$. Thus, an increasing trend was

observed in all three treatments (Figure 1). $N_2O$ accumulated towards the end of the incubations in treatments OD and AD,

with concentrations above 2.0 mmol/L. $^{45}N_2O$ and $^{46}N_2O$ evolution also provided evidence of $N_2O$-denitrification during the

S2 stage, both increasing over time (Figure 2). However, once the original $NO_3^-$ had been consumed, increases in both $^{45}N_2O$

and $^{46}N_2O$ for the oxic treatments can be attributed to other processes such as $^{15}N$ recirculation by coupled DNRA-

nitrification (DNRA fueling nitrification to $N_2O$), a process whose importance has recently been highlighted in estuarine

sediments (Dunn et al., 2009; Murphy et al., 2016). Finally, the solubility of $N_2O$ at 50 g/L of salinity and 25 ºC was 14.25

mmol/L, whereas the solubility for $N_2$ at the same conditions was significantly lower (0.43 mmol/L). Therefore, $N_2$

oversaturation is observed in the water column (Figure 1), clear evidence for the presence of the above-described $NO_3^-$

reduction processes (Wenk et al., 2013; 2014). Concentrations of $N_2$ remained almost constant throughout the incubations ($\approx$

6 mmol/L), regardless of whether atmospheric air (OL and OD) or argon (AD) was bubbling in the mesocosm. Thus, $N_2$

oversaturation is probably the result of N reduction and further accumulation in the water column, which has not yet reached

atmospheric equilibrium for nitrogen (Weiss and Craig, 1973). About $^{29}N_2$ and $^{30}N_2$ concentrations, used for the IPT

calculations, they remained almost unchanged and below the maximum solubility value (Figure 2). In addition, the $^{15}N$ mass

balance was calculated to detect whether gas bubbling (atmospheric air or argon to maintain mesocosms in an aerobic or





anoxic state) and differences in solubility may strip $^{29}N_2$ and $^{30}N_2$ faster than $^{45}N_2O$ and $^{46}N_2O$ (Figure 3). Mean $^{15}N$ recoveries were 92% for OL (from 79 to 108%), 94% for OD (from 67 to 125%), and 93% for AD (from 73 to 126 %). $^{15}N$ losses of 6-8% based on whole-system $^{15}N$ recoveries seem very small and may derive mainly from the accumulation of

errors in the measurements (concentrations and at%$^{15}N$ enrichments) of 4-5 dissolved and gaseous N pools. Therefore, we consider that there were no significant N losses deriving from gas bubbling, but if so the experiment would have only underestimated $N_2$ production processes (i.e. full denitrification and anammox).

Figure 4 shows the evolution of physico-chemical parameters. The evolution of DOC and DON in the water column showed stable concentrations during S1, followed by a sharp increase in S2 after tracer addition (Figure 4). Diatom blooms can

disrupt the ecological balance, causing the breakdown of cyanobacterial populations, and the release of large amounts of dissolved OM (Xue et al., 2017). Afterwards, DOC and DON decreased as a result of heterotrophic metabolism. In S3, we explain the observed DOC changes first by consumption and finally by accumulation after bloom collapse. DON increases are likely related to phytoplankton decay with subsequent organic N mineralization. Decreasing percentages of DON:DNb underline the role of OM remineralization throughout the incubation. Compared to initial conditions, pH decreased in OL,

OD and AD treatments. The decrease of pH values is probably due to the release of organic acids and $CO_2$, both produced from carbon sources during microbial metabolism. pH values were within the optimal range for denitrification (Knowles, 1982) and DNRA (Van den Berg et al., 2015). Eh abruptly dropped in S1 and maintained negative values during S2 and S3, with a small rise after tracer addition (Figure 4).

### 3.3 Nitrous oxide production

The contribution of each process to total N removal was calculated for each mesocosm and treatment. ANOVA results provided evidence of a (co-)dominant role of $N_2O$-denitrification in OL and OD treatments, with 82% and 81% of N removal, respectively (Figure 5). Contribution of $N_2O$-denitrification to total N loss was significantly higher than reported for aquatic sediments (< 8.6%; Risgaard-Petersen et al., 2003; McCrackin and Elser, 2010). Treatments OL and OD showed mean $N_2O$-denitrification rates of 1.76 ($\pm$ 2.58) and 2.10 ($\pm$ 2.24) mmol N m$^{-2}$ h$^{-1}$, respectively. Such high values have been

reported previously only in tropical wetland soils (up to 1.56 mmol N m$^{-2}$ h$^{-1}$; Liengaard, et al., 2014) and estuarine sediments affected by agricultural activities (up to 4.85 mmol N m$^{-2}$ h$^{-1}$; Salahudeen et al., 2018) (Table 4). These results support evidence from previous observations (Huttunen et al., 2003), which showed that lakes subjected to elevated N inputs are an important source of $N_2O$ emissions. However, studies involving the role of $N_2O$-denitrification in saline aquatic environments are mainly restricted to marine ecosystems. Our high measured rates may be explained by the high biological

activity after $^{15}N$-$NO_3^-$ addition, in the absence of nutrient limitation. Nonetheless, the $N_2O$ production in field studies is most probably limited by N availability. Another potential source of $N_2O$ production is partial nitrification. In treatments OD and OL, the conditions are met for this process to take place. No $NO_3^-$ was measured after the tracer addition and further consumption, but $N_2O$ was produced over time. In fact, Figure 6 shows that $N_2O$-denitrification was the main pathway in the





S3 stage for such treatments, which may mask the contribution of nitrification fueled by DNRA. Finally, abiotic contribution

to $N_2O$ production may also contribute to produce $N_2O$ in hypersaline environments (Samarkin et al., 2010).

The AD treatment showed a similar average value of $N_2O$-denitrification ($1.87 \pm 3.99$ mmol N m$^{-2}$ h$^{-1}$) than treatments OL and OD, being similar to rates reported for pristine mangrove sediments (up to 0.67 mmol N m$^{-2}$ h$^{-1}$; Fernandes et al., 2010), but higher rates of $N_2$-denitrification than OL and OD. Therefore, $N_2O$-denitrification showed a smaller yet still dominant contribution to total N removal in the AD treatment. A possible explanation for this pattern is that $N_2O$ reductase activity is

sensitive towards oxygen (Bonin and Gilewicz, 1991), being partially inhibited in treatments OL and OD in the presence of dissolved $O_2$ (~ 6.4 mg/L in the water column), thereby decreasing $N_2$-denitrification in aerated treatments. Overall, $N_2O$-denitrification showed a significant contribution to $NO_3^-$ reduction during the whole sediment incubations together with DNRA (Figure 6). In terms of $NO_3^-$ reduction, when $N_2O$-denitrification was of greater importance, DNRA and anammox showed a smaller contribution to $NO_3^-$ reduction, and vice versa. As suggested above, the influence of partial nitrification as

a source of $N_2O$ cannot be discarded. Unfortunately, additional experiments would have been necessary which were not the main focus of this study (pathways of nitrate removal).

$N_2$-denitrification showed the highest rates at the beginning of the incubation (AD, $\leq 13$ mmol N m$^{-2}$ h$^{-1}$). Mean measured $N_2$ production rates attributed to denitrification in the OL treatment was 0.05 mmol N m$^{-2}$ h$^{-1}$, in accordance with intact estuarine sediments (0.036 – 0.155 mmol N m$^{-2}$ h$^{-1}$; Trimmer et al., 2003) and contributed on average 4% to total N removal

(Figure 3). $N_2$-denitrification played a greater role in $NO_3^-$ reduction under darkness: 11% and 13% of the total N removal in OD and AD treatments, respectively These results agree with earlier observations (Risgaard-Petersen et al., 1994) which showed reduced denitrification rates associated with light exposure and photosynthesis by benthic microphytes. In the OD treatment, mean production rate was 0.41 ($\pm 1.57$) mmol N m$^{-2}$ h$^{-1}$ by $N_2$-denitrification (Table 3). Our values were similar to those reported in marine environments like Heron Island (0.48 mmol N m$^{-2}$ h$^{-1}$; Eyre and Ferguson, 2008) and Randers Fjord

(0.34 mmol N m$^{-2}$ h$^{-1}$; Risgaard-Petersen et al., 2004) (Table 4). Highest $N_2$-denitrification rates were found in the AD treatment with an average value of 0.80 ($\pm 2.61$) mmol N m$^{-2}$ h$^{-1}$. These results were close to those reported by Erler et al. (2008) (0.652 – 0.966 mmol N m$^{-2}$ h$^{-1}$), where denitrifiers coexisted with anammox bacteria in a constructed wetland which received secondary treated sewage effluents. The largest contribution of $N_2$-denitrification was detected at the initial stages of incubation, coupled to higher DOC concentrations, but also during late stages of incubation in OD (~ 30%) (Figure 4).

These results suggest the dominance of heterotrophic denitrification linked to the breakdown of biomass (Xue et al., 2017).

### 3.4 Close coupling between DNRA and anammox

Total N removal and $NO_3^-$ reduction reached highest values under anoxia and darkness conditions (mean of $3.63 \pm 5.30$ mmol N m$^{-2}$ h$^{-1}$ and $6.43 \pm 6.56$ mmol N m$^{-2}$ h$^{-1}$, respectively; Table 3). As discussed above, under those conditions DNRA was the dominant process. These results are consistent with hydrochemical data, which showed a significant accumulation of

N-$NH_4^+$ in the water column in the AD treatment.





Previous research showed favorable conditions for DNRA activity in sediments from Pétrola Lake, such as high organic C:N ratios or the presence of microorganisms capable of performing DNRA (Valiente et al., 2017; Valiente et al., 2018). Average DNRA rates in OL and OD treatments (~ 1.4 mmol N m$^{-2}$ h$^{-1}$) were similar to those reported for anoxic estuarine sediments where DNRA was the dominant process (1.140 mmol N m$^{-2}$ h$^{-1}$; Dong et al., 2011). In the AD treatment, mean DNRA rates

(2.80 ± 2.56 mmol N m$^{-2}$ h$^{-1}$) were similar to those observed in nutrient enriched environments like fringing wetlands (up to 6.13 mmol N m$^{-2}$ h$^{-1}$; Tobias et al., 2011) or eutrophic shelf seas (up to 3.58 mmol N m$^{-2}$ h$^{-1}$; Song et al., 2013) (Table 4). In line with the data obtained in the ANOVA tests, the $^{15}$N-IPT data showed that NO$_3^-$ reduction by DNRA was significantly higher in AD (52%) than in OL (41%) and OD (35%) (Figure 5). The contribution of DNRA was in the same range as reported for estuarine and salt marsh sediments (Dong et al., 2009; Koop-Jakobsen and Giblin, 2010), fostering the retention

of reactive nitrogen in the system. Compared to denitrification, DNRA contributed more to NO$_3^-$ reduction after the initial incubation phase, approximately from time 2.5 h onwards (Figure 4). Recent studies also demonstrated that DNRA is stimulated in the presence of H$_2$S at the expense of denitrification (Roland et al., 2017). Our results support those findings: AD provided the most favorable conditions for bacterial sulfate-reduction (Table 2), and H$_2$S production in Pétrola sediments (Valiente et al., 2017) can reach values up to 0.024 nmol/cm$^3$·s.

DNRA therefore plays the key role in increasing N-NH$_4^+$ contents in the water column, being more relevant than OM remineralization and sedimentary release. Existing NH$_4^+$ may be oxidized to NO$_2^-$ both under aerobic and anaerobic conditions (Schmidt et al., 2002), contributing to a temporary increase of N-NO$_2^-$ and promoting NO$_2^-$ and NH$_4^+$ consumption by anammox bacteria. Moreover, N-NH$_4^+$ release does fuel N loss from the system via coupled DNRA-anammox. Therefore, DNRA and anammox bacteria acting together may have an energetic advantage over denitrifiers in the

competition for substrates under low oxygen conditions (Jensen et al., 2011). The close reliance of anammox on DNRA has been reported in marine ecosystems with high N loss via anammox, mainly linked to the availability of OM (Kalvelage et al., 2013). In Pétrola Lake sediment incubations, anammox seems to be fueled by DNRA. This interpretation is based on the similar trend in the contribution of both processes to total NO$_3^-$ reduction (AD>OL>OD; Figure 3). Coupled DNRA-anammox showed a higher contribution in all treatments than canonical anammox (Table 3), corroborating the key role of

DNRA in fueling N loss pathways.

The isotope data clearly confirm the presence of anammox (Table 3). The mean rates of N loss via anammox in OL and OD treatments (~0.4 mmol N m$^{-2}$ h$^{-1}$) were in the range of previous studies in eutrophic sediments (up to 0.413 mmol N m$^{-2}$ h$^{-1}$; Han and Li, 2016), but significantly lower than those found in the AD treatment (0.96 mmol N m$^{-2}$ h$^{-1}$). These results agree with recent studies showing the importance of anammox activity in the presence of H$_2$S in freshwater lakes (Roland et al.,

2017), conditions which are given for the highly saline lake studied here. On average, the contribution of anammox to total N loss ranged from 8% (OD) to 28% (AD) (Figure 3). This range corresponds with studies performed in continental shelf sediments (Song et al., 2013) (28%), intertidal sediments (Hsu and Kao, 2013) (12%), and is close to the global mean value including inland waters (Trimmer and Engström, 2011) (23%). Finally, the higher the participation of anammox in total N

removal, the lower the relevance of $N_2O$-denitrification, leading to decreased $N_2O$ emissions from hypersaline lake
ecosystems.

## 4 Conclusions

The purpose of the current study was to determine the influence of oxygen and light in the water column on the balance
between nitrate removal pathways during incubations of lacustrine organic-rich sediments. Our findings provide evidence for
the coexistence of denitrification, DNRA, and anammox in a highly saline lake. The application of the revised [15]N-IPT
highlighted the importance of coupled DNRA-anammox during our incubations. We showed here that DNRA and $N_2O$-
denitrification played a predominant role in N removal under the studied conditions, showing unexpectedly high $N_2O$
emission rates compared to previous studies. However, DNRA was the key process when oxygen and light were absent from
the water column. Then, anammox also had a greater influence on total N removal with markedly high rates (up to 0.96
mmol N m$^{-2}$ h$^{-1}$). It seems that anoxia and darkness promoted DNRA, the critical process which fuels anammox. As a result,
these conditions reduced $N_2O$ emissions to the atmosphere. As far as we know, the role of coupled DNRA-anammox in such
saline ecosystems has not yet been explored, and therefore, anammox was typically underestimated. Further research is
required to fully understand the role of coupled DNRA-anammox in N cycling in lake ecosystems, as well as the influence
that coupled DNRA-nitrification can exert on $N_2O$ production.

## Acknowledgements

The work was supported by a PhD grant (BES-2012-052256) and project CICYT- CGL2017-87216-C4-2-R from the
Spanish government, the SBPLY/17/180501/000296 project from the Castilla–La Mancha regional government, and funds
for a Research Visit to Vienna (UCLM). We thank M. Stachowitsch for English copyediting and valuable comments. Special
thanks to A. Menchén, B. Toledo, M. A. Gutiérrez, A. Valenciano, and A. García for their laboratory help. The authors are
also grateful for the analytical assistance to all colleagues from the Environmental and Radiochemistry Group (University of
Vienna), from the SILVER Lab (University of Vienna), from WasserCluster Lunz, and from the MEB Group (Aix-Marseille
Université), and to an anonymous reviewer for helpful comments and suggestions.

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





**Figure 1: Inorganic N species evolution over time in the three studied treatments. Nitrate (NO₃⁻), nitrite (NO₂⁻), ammonium (NH₄⁺), dinitrogen gas (N₂) and nitrous oxide (N₂O) evolution over time. Three different stages could be distinguished during incubations: a first stage (S1) extended from sampling (NC₋₄₈) to ¹⁵N-NO₃⁻ addition (time 0 h), the second stage (S2) covered the time between 0 and 24 h, and the third stage (S3) from time 24 h to the end of incubation (time 72 h). The negative times indicate the stabilization period. ¹⁵N-NO₃⁻ was added at 0 h. S1, S2, and S3 refer to the three stages distinguished during incubations. Error bars represent ± 1 standard deviation.**



**Figure 2: $^{15}$N evolution over time.** Evolution of $^{15}NH_4^+$, $^{15}NO_3^-$, $^{29}N_2$, $^{30}N_2$, $^{45}N_2O$ and $^{46}N_2O$ concentration from the $^{15}$N-NO$_3^-$ addition. It includes second stage (S2; 0 - 24 h) and third stage data (S3; 24 - 72 h).




**Figure 3: Mass balance of $^{15}$N during incubation time in the three treatments. Percentage of $^{15}$N recovery in $^{15}$NH$_4^+$, $^{15}$NO$_3^-$, $^{15}$N$_2$ (including $^{29}$N$_2$ and $^{30}$N$_2$), and $^{15}$N$_2$O (including $^{45}$N$_2$O and $^{46}$N$_2$O) at each time point for the three treatments of the study.**




**Figure 4: Physical-chemical, DOC, and DON evolution over time. pH, redox potential (Eh), dissolved oxygen (DO), dissolved organic carbon (DOC), and dissolved organic nitrogen (DON) evolution over time in the three studied treatments (OL, OD, and**

**AD). The negative times indicate the stabilization period. Error bars represent ± 1 standard deviation.**



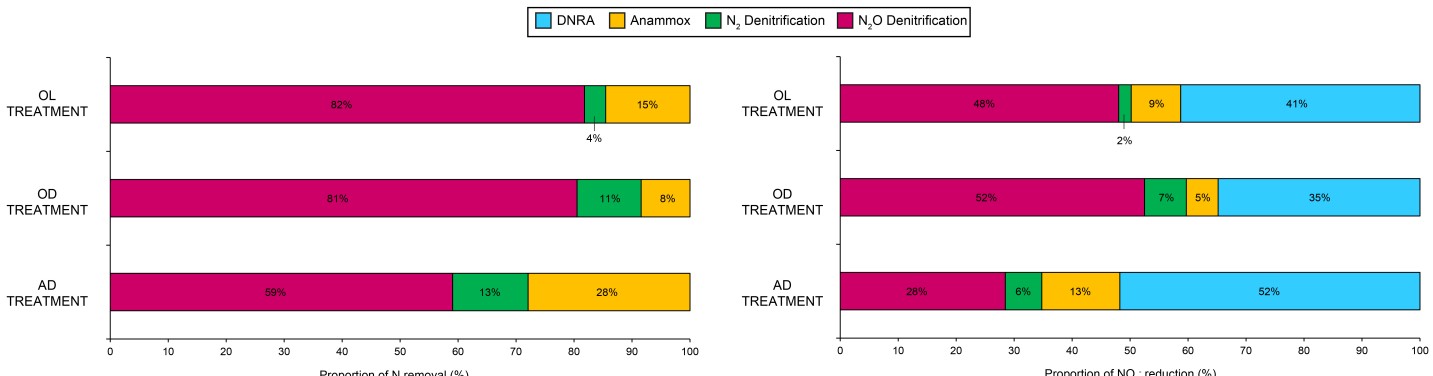

**Figure 5: Contribution of each pathway to total nitrogen removal and nitrate reduction. Proportion of N₂-denitrification, N₂O-denitrification, and anammox to total N removal (left). Contribution of DNRA, N₂-denitrification, N₂O-denitrification, and anammox to nitrate reduction (right). Rates were measured at three different incubation conditions (treatments OL, OD, and AD).**



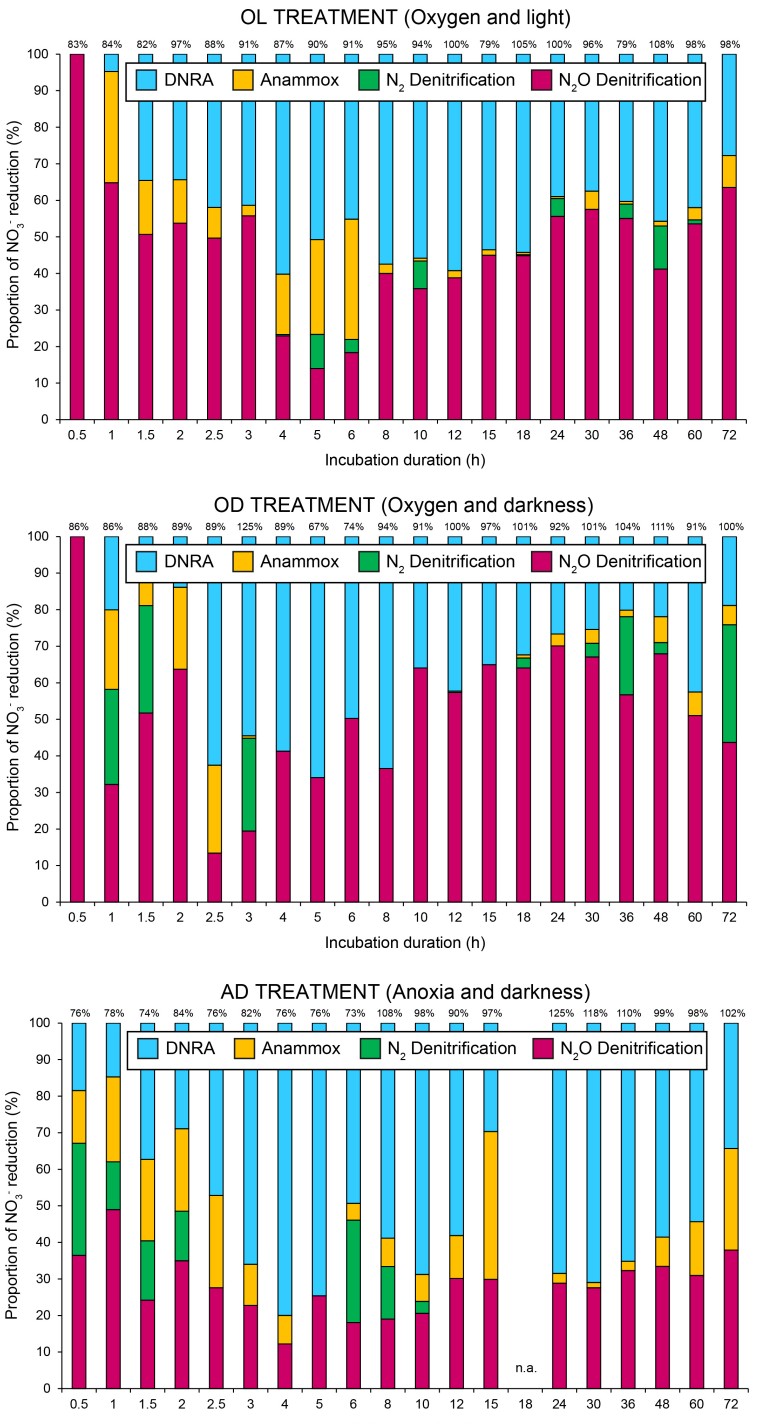

**Figure 6: Evolution of the contribution to nitrate reduction processes over time. Proportion of N₂-denitrification, N₂O-denitrification, DNRA, and anammox over time under three incubation conditions (treatments OL, OD, and AD). Proportions of each process were measured at twenty different incubation times, by triplicate per treatment. Recovery percent of the initial ¹⁵N at each time is shown above the bars. Detailed mass balance is included in Figure S2. n.a.: not available.**





**Table 1. Parameters and equations used in the $^{15}$N-IPT modeling based on Salk et al. (2017).**

| Parameter | Description | Equation |
|---|---|---|
| $P_{29}$ | $^{29}N_2$ production (measured directly) | |
| $P_{30}$ | $^{30}N_2$ production (measured directly) | |
| $P_{45}$ | $^{45}N_2O$ production (measured directly) | |
| $P_{46}$ | $^{46}N_2O$ production (measured directly) | |
| $r_{14}$ | $^{14}N/^{15}N$ ratio in $NO_3^-$ (measured directly) | |
| $r_{14a}$ | $^{14}N/^{15}N$ ratio in $NH_4^+$ (measured directly) | |
| $P_{15-NH4}$ | $^{15}NH_4^+$ production (measured directly) | |
| $D_{28}$ | $^{28}N_2$ production via denitrification | $r^2_{14} \cdot (P_{30} - A_{30})$ |
| $D_{29}$ | $^{29}N_2$ production via denitrification | $2r_{14} \cdot (P_{30} - A_{30})$ |
| $D_{30}$ | $^{30}N_2$ production via denitrification | $P_{30} - A_{30}$ |
| $A_{28}$ | $^{28}N_2$ production via anammox | $(r_{14} \cdot A_{29}) / [(r_{14}/r_{14a})+1]$ |
| $A_{29}$ | $^{29}N_2$ production via anammox | $P_{29} - D_{29}$ |
| $A_{29-DNRA}$ | $^{29}N_2$ production via coupled DNRA-anammox | $A_{29} / [1 + (r_{14}/r_{14a})]$ |
| $A_{29-Anammox}$ | $^{29}N_2$ production via canonical anammox | $A_{29} - A_{29-DNRA}$ |
| $A_{30}$ | $^{30}N_2$ production via anammox | $(P_{29} - 2P_{30} \cdot r_{14}) / (r_{14a} - r_{14})$ |
| $D_{14}$ | $N_2$ genuine production via denitrification | $2D_{28} + D_{29}$ |
| $D_{14 -N2O}$ | $N_2O$ genuine production via denitrification | $r_{14} \cdot (2P_{46} + P_{45})$ |
| $A_{14}$ | $N_2$ genuine production via anammox | $2A_{28} + A_{29-DNRA}$ |
| $A_{14 -DNRA}$ | $N_2$ genuine production via coupled DNRA-anammox | $A_{29-DNRA} + 2A_{28} \cdot (A_{29-DNRA} / A_{29})$ |
| $A_{14 -Anammox}$ | $N_2$ genuine production via canonical anammox | $2A_{28} \cdot (A_{29-Anammox} / A_{29})$ |
| $P_{14}$ | Total $N_2$ + $N_2O$ genuine production | $D_{14} + A_{14} + D_{14-N2O}$ |
| $DNRA$ | $NH_4^+$ genuine production via DNRA | $r_{14} \cdot (P_{15-NH4} + A_{30})$ |





**Table 2. Mean values (±SD) of physico-chemical parameters in water and sediment for the experiments at the beginning and at the end of incubations.**

| Treatment | Conditions water column | pH | Eh (mV) | DO (mg/L) | EC (mS/cm) | TDS (g/L) | DOC (mmol $L^{-1}$) | DNb (mmol $L^{-1}$) | DON (mmol $L^{-1}$) |
|---|---|---|---|---|---|---|---|---|---|
| $NC_{-48}$ (*) | Natural conditions (lake) | 8.70 | +135.1 | 4.26 | 72.1 | 45.1 | 16.3 | 1.07 | 1.00 |
| $OL_{72}$ (n=3) | Aeration and light | 7.89 ± 0.08 | -105.7 ± 28.1 | 6.46 ± 0.26 | 75.9 ± 3.90 | 45.5 ± 3.30 | 27.0 ± 2.9 | 1.12 ± 0.17 | 0.98 ± 0.16 |
| $OD_{72}$ (n=3) | Aeration and darkness | 7.93 ± 0.13 | -114.3 ± 41.0 | 6.40 ± 0.73 | 76.8 ± 0.86 | 47.1 ± 0.56 | 31.3 ± 2.6 | 1.30 ± 0.15 | 1.12 ± 0.15 |
| $AD_{72}$ (n=3) | Anoxia and darkness | 7.40 ± 0.35 | -370.7 ± 7.51 | 0.08 ± 0.02 | 80.9 ± 0.91 | 50.1 ± 0.80 | 28.7 ± 5.7 | 1.21 ± 0.08 | 1.01 ± 0.09 |

| Treatment | DON:DNb (%) | N-NO3 (µmol $L^{-1}$) | N-NH4 (µmol $L^{-1}$) | N-NO2 (µmol $L^{-1}$) | N2 (mmol $L^{-1}$) | N2O (mmol $L^{-1}$) | LOI (%) | S-N-NO3 (µmol $kg^{-1}$) | S-N-NH4 (mmol $kg^{-1}$) | S-N-NO2 (µmol $kg^{-1}$) |
|---|---|---|---|---|---|---|---|---|---|---|
| $NC_{-48}$ (*) | 93.2 | 9.25 | 63.1 | BLD | n.a. | n.a. | 8.50 ± 2.06 | 17.1 ± 2.78 | 1.28 ± 0.37 | BLD |
| $OL_{72}$ (n=3) | 87.5 ± 0.5 | BLD | 139 ± 15.7 | BLD | 6.07 ± 0.28 | 0.52 ± 0.05 | 8.84 ± 1.56 | 68.4 ± 13.4 | 0.60 ± 0.27 | BLD |
| $OD_{72}$ (n=3) | 86.4 ± 1.7 | BLD | 175 ± 10.9 | BLD | 5.88 ± 0.02 | 2.28 ± 2.69 | 8.51 ± 1.46 | 64.6 ± 24.4 | 1.22 ± 0.76 | BLD |
| $AD_{72}$ (n=3) | 83.5 ± 2.0 | BLD | 198 ± 12.2 | BLD | 6.27 ± 0.74 | 2.23 ± 3.09 | 9.90 ± 1.31 | 73.2 ± 16.0 | 1.83 ± 0.27 | BLD |

(*) At $NC_{-48}$: n = 1 in water samples for determination of all the chemical parameters; n = 3 in sediment samples (LOI, $S-N-NO_3^-$, $S-N-NH_4^+$, and $S-N-NO_2^-$). Eh: redox potential. DO: dissolved oxygen. EC: electrical conductivity. TDS: total dissolved solids. DOC: dissolved organic carbon. DNb: dissolved bound nitrogen. DON: dissolved organic nitrogen. LOI: loss of ignition. BLD: below limit of detection. n.a.: not available.





**Table 3. Mean (±SD) and maximum rates of N-loss processes after 72 h of mesocosm incubations.**

N-conversion rates (mmol N m$^{-2}$ h$^{-1}$)

| Mesocosms | Total N removal | | Total NO$_3^-$ reduction | | N$_2$-Denitrification | | N$_2$O-Denitrification | | DNRA-Anammox | | Canonical Anammox | | N$_2$-Anammox | | DNRA | |
|---|---|---|---|---|---|---|---|---|---|---|---|---|---|---|---|---|
| | mean | max | mean | max | mean | max | mean | max | mean | max | mean | max | mean | max | mean | max |
| OL-1 (n=20) | 2.22 (± 2.13) | 6.88 | 3.72 (± 3.26) | 10.5 | 0.13 (± 0.25) | 0.85 | 1.85 (± 1.92) | 6.89 | 0.17 (± 0.34) | 1.10 | 0.07 (± 0.12) | 0.42 | 0.24 (± 0.46) | 1.52 | 1.50 (± 1.57) | 5.01 |
| OL-2 (n=20) | 2.16 (± 3.42) | 12.3 | 3.46 (± 4.35) | 15.9 | 0.00 (± 0.02) | 0.07 | 1.39 (± 3.13) | 12.3 | 0.55 (± 0.86) | 2.49 | 0.22 (± 0.40) | 1.26 | 0.77 (± 1.24) | 3.38 | 1.29 (± 1.25) | 3.65 |
| OL-3 (n=20) | 2.29 (± 2.62) | 9.78 | 4.09 (± 3.92) | 15.1 | 0.03 (± 0.11) | 0.46 | 2.03 (± 2.69) | 9.78 | 0.15 (± 0.45) | 1.66 | 0.09 (± 0.25) | 0.78 | 0.24 (± 0.70) | 2.45 | 1.80 (± 1.76) | 5.59 |
| **OL (n=60)** | **2.23 (± 2.71)** | **12.3** | **3.76 (± 3.79)** | **15.9** | **0.05 (± 0.16)** | **0.85** | **1.76 (± 2.58)** | **12.29** | **0.29 (± 0.61)** | **2.49** | **0.12 (± 0.28)** | **1.26** | **0.41 (± 0.88)** | **3.38** | **1.54 (± 1.53)** | **5.59** |
| OD-1 (n=20) | 1.54 (± 2.10) | 7.65 | 2.30 (± 2.10) | 7.65 | 0.10 (± 0.28) | 1.00 | 1.43 (± 2.15) | 7.65 | 0.00 (± 0.01) | 0.05 | 0.00 (± 0.01) | 0.04 | 0.01 (± 0.02) | 0.09 | 0.76 (± 0.79) | 2.01 |
| OD-2 (n=20) | 2.54 (± 2.56) | 11.0 | 4.33 (± 3.36) | 15.4 | 0.08 (± 0.17) | 0.70 | 1.70 (± 1.55) | 5.31 | 0.41 (± 0.88) | 3.62 | 0.36 (± 0.87) | 3.34 | 0.77 (± 1.72) | 6.96 | 1.79 (± 1.36) | 4.41 |
| OD-3 (n=20) | 4.35 (± 4.73) | 14.7 | 5.73 (± 4.92) | 16.7 | 1.03 (± 2.59) | 8.33 | 3.07 (± 2.68) | 9.81 | 0.14 (± 0.57) | 2.48 | 0.11 (± 0.47) | 2.05 | 0.25 (± 1.04) | 4.53 | 1.38 (± 1.16) | 3.94 |
| **OD (n=60)** | **2.87 (± 3.51)** | **14.7** | **4.22 (± 3.89)** | **16.7** | **0.41 (± 1.57)** | **8.33** | **2.10 (± 2.24)** | **9.81** | **0.20 (± 0.64)** | **3.62** | **0.17 (± 0.60)** | **3.34** | **0.37 (± 1.22)** | **6.96** | **1.35 (± 1.20)** | **4.41** |
| AD-1 (n=20) | 4.52 (± 6.00) | 21.0 | 7.41 (± 8.12) | 25.3 | 0.00 (± 0.00) | 0.00 | 4.40 (± 6.00) | 21.0 | 0.08 (± 0.30) | 1.33 | 0.04 (± 0.15) | 0.66 | 0.12 (± 0.45) | 1.99 | 2.89 (± 3.12) | 13.9 |
| AD-2 (n=20) | 3.18 (± 5.64) | 19.7 | 6.16 (± 6.14) | 22.1 | 1.79 (± 3.93) | 13.0 | 0.40 (± 1.12) | 4.96 | 0.66 (± 1.51) | 5.73 | 0.33 (± 0.66) | 1.84 | 0.99 (± 2.12) | 7.54 | 2.98 (± 2.21) | 8.68 |
| AD-3 (n=20) | 3.16 (± 4.21) | 13.5 | 5.68 (± 5.28) | 19.7 | 0.58 (± 1.95) | 8.17 | 0.75 (± 1.07) | 3.97 | 0.80 (± 1.37) | 5.36 | 1.02 (± 2.26) | 7.26 | 1.82 (± 3.48) | 11.0 | 2.53 (± 2.36) | 8.64 |
| **AD (n=60)** | **3.63 (± 5.30)** | **21.0** | **6.43 (± 6.56)** | **25.3** | **0.80 (± 2.61)** | **13.0** | **1.87 (± 3.99)** | **21.0** | **0.51 (± 1.21)** | **5.73** | **0.46 (± 1.38)** | **7.26** | **0.96 (± 2.40)** | **11.0** | **2.80 (± 2.56)** | **13.9** |





**Table 4. Published rates of sedimentary denitrification, DNRA and anammox measured in intact sediment cores (mmol N m$^{-2}$ h$^{-1}$). n.a.: not available**

| Source | DNRA | Anammox | N$_2$-Denitrification | N$_2$O-Denitrification | Reference |
|---|---|---|---|---|---|
| Pétrola Lake (Spain) | 0 - 2.800 | 0 - 0.960 | 0 - 0.800 | 0 - 2.100 | This study |
| Colne estuary (United Kingdom) | 0.005 - 0.400 | 0.157 | n.a. | n.a. | Dong et al. (2009) |
| Cisadane estuary (Indonesia) | 1.140 | n.a. | n.a. | n.a. | Dong et al. (2011) |
| Thau lagoon (France) | 6.708 | n.a. | n.a. | n.a. | Gilbert et al. (1997) |
| East China Sea shelf (China) | 0.791 - 3.583 | n.a. | n.a. | n.a. | Song et al. (2013) |
| Fringing marsh-aquifer ecotone (USA) | 0.875 - 6.125 | n.a. | 1.800 - 17.60 | n.a. | Tobias et al. (2001) |
| Plum Island Sound estuary (USA) | 0.004 - 0.310 | n.a. | 0 - 0.332 | n.a. | Koop-Jakobsen and Giblin (2010) |
| German Bight (Germany) | 0.010 | n.a. | 0.124 | n.a. | Marchant et al. (2016) |
| Heron Island (Australia) | n.a. | n.a. | 0.034 - 0.480 | n.a. | Eyre and Ferguson (2008) |
| Lake Tanganyika (Burundi, DRC, Tanzania, Zambia) | n.a. | 0.100 | n.a. | n.a. | Schubert et al. (2006) |
| Randers Fjord (Denmark) | n.a. | 0.014 - 0.021 | 0.219 - 0.335 | n.a. | Risgaard-Petersen et al. (2004) |
| Thames estuary (United Kingdom) | n.a. | 0 - 0.010 | 0.036 - 0.155 | n.a. | Trimmer et al. (2003) |
| Gravesend, Thames estuary (United Kingdom) | n.a. | 0.049 | 0.193 | n.a. | Trimmer et al. (2006) |
| Constructed wetland in New South Wales (Australia) | n.a. | 0.066 - 0.199 | 0.652 - 0.966 | n.a. | Erler et al. (2008) |
| Taihu Lake (China) | n.a. | 0.049 - 0.413 | 0.132 - 0.656 | n.a. | Han and Li (2016) |
| Lake Superior (Canada, USA) | n.a. | 0.021 - 0.040 | 0.019 - 0.128 | n.a. | Crowe et al. (2017) |
| Danshuei estuary (Taiwan) | n.a. | 0.013 | 0.126 | 0.050 | Hsu and Kao (2013) |
| Pearl River estuary (China) | n.a. | 0 - 0.003 | 0.032 - 0.708 | 0 - 0.022 | Tan et al. (2019) |
| Lake Bonney (Antarctica) | n.a. | n.a. | n.a. | 0.191 | Prisu et al. (1996) |
| Tuvem and Divar, Goa (India) | n.a. | n.a. | n.a. | 0.140 - 0.670 | Fernandes et al. (2010) |
| Pantanal wetland (Brazil) | n.a. | n.a. | n.a. | 0 - 1.560 | Liengaard et al. (2014) |
| Ashtamudi estuary (India) | n.a. | n.a. | n.a. | 0.490 - 4.850 | Salahudeen et al. (2018) |