# Peer review of "Oxygen and light determine the pathways of nitrate reduction in a highly saline lake"

_Biogeosciences, 2020_

## Referee Comment (RC1) · Anonymous Referee #1 · 12 Feb 2020

Review of Valiente et al for BGS

The authors aim to determine rates of denitrification, anammox and DNRA in sediments of a shallow saline lake heavily impacted by anthropogenic N inputs. The study aims to investigate the impact of light and/or oxygen conditions on the balance between nitrate reduction processes. This is one of the few studies which have been able to use the 'new' IPT methods accounting for the co-ocurrence of the three nitrate reducing processes which is to the authors' merit. Indeed the authors demonstrate the importance of coupled DNRA-anammox in the sediments and how anammox would be underestimated through $30N_2$ production. However, in order to accept the manuscript it needs a major revision of structure in all sections, streamlining of the text, review of data/figures and thorough proof-reading before resubmission.

*Major general comments:*

As above, please re-consider the structure of how each section is approached and should be proof-read. I also suggest it's less confusing to keep Results and discussion as separate sections.

Check nitrate/$NO_3^-$ nitrite/$NO_2^-$ throughout for consistency

Consider the relevance of references you use… some are from water column (e.g. Jensen et al 2011) or sediment with very difference settings.

*Title:* Add in 'sediment' somewhere

*Introduction*

The introduction is quite disjointed with no logical direction to draw the reader in. For future potential submissions I would suggest a nicer structure to introduce the reader to your topic – e.g.:

- Shortly introduce N cycle and identify large anthropogenic impacts
- Overview on N cycling processes **in sediments**, focus on nitrate reducing processes/end products
- Introduce saline lakes and their importance/why are they interesting/understudied
    - Potential factors controlling NO3- reducing processes in this lake (light, O2 etc)

(I should say that I didn't realise the study was only on sediments until line 80 as this in not explicitly mentioned before. Additionally, Jensen et al 2011 is a water column study so might not be relevant to sediments (where there should anyway be plenty of NH4+)).

*Methods*

Please re-think the structure and need for the amount of text here

A lot of the text in the Methods section is unnecessary. If you are referencing a method from another paper, it should only be very briefly described in your methods (e.g. a lot of things don't need to be reiterated from Salk et al).

What is the relevance of the treatments to your study site? Does the lake become stratified/anoxic in some months? What is the phytobenthos? Diatoms? Bacterial mats? This needs to be put into context with better descriptions of your site (in methods and results)

Section 2.1:What depth were water samples taken?

Nutrient samples should be filtered (at least 0.2, possibly 0.45um filters) so no nutrients are produced/consumed between sampling/measuring.

Welti et al use a reservoir to feed the sealed (gas-space-free) mesocosms, I'm very confused about the method description here, was the mesocosm water itself bubbled?

Section 2.2:

The first paragraph is a very long way to write 'the overlying water of each mesocosm was bubbled with either air (oxic treatment) or Argon (anoxic treatment)

Why do you seal mescosms with no air space if you're going to bubble them anyway?

Line 135: what do you mean by pump? Do you mean a wheel/stirrer to mix the water to avoid stagnation? If not please explain more clearly – and add how the mesocosm water was mixed.

Was light intensity monitored/measured?

It's fine to use Dalsgaard et al 2000 for timing calculations and assuming 1mm oxygen penetration. However an oxygen penetration depth of 1mm is not correct for the anoxic treatments.

Why was this very high resolution time series chosen? More reasoning should be presented behind this as it seems a bit unnecessary. What is the relevance of 24h anoxia and darkness in a 2m deep lake? Diel and seasonal conditions should be better described in terms of biogeochemistry and phytopenthos etc.

Was water removed though sampling the mesocosms replaced? Were dilution effects accounted for?

In the anoxic treatment why do you assume all $NO_3^-$ reduction takes place in the sediment? Why not also the water column?

Line 164-165: Is this the name of the site? This has not been introduced/mentioned until now.

Section 2.4: A lot of this text is unnecessary and can be streamlined.

Line 181: Use original references for the microdiffusion method.

*Results and Discussion* (I suggest it's better to split into two sections)

This section is very confusing to read so please consider re-structuring. All of 3.1 seems to be results and 3.2 onwards is a mixture of results/discussion.

It's also important to include the *in situ* conditions you measured and a description of the site. What is the relevance of a longer anoxic incubation to your site?

What are the subscript numbers? E.g. $OL_{72}$, $t_{(2)}$, $F_{(2,6)}$

Don't forget units (e.g. lines 262-270)

In general, a more thorough discussion of data is needed.

The 'stages' 1, 2 and 3 have not been defined/introduced until the results, please re-consider how you refer to the experiments.

Why is there already so much 15N-$N_2$ at the time when the tracer is added (in Fig 2)? I suggest the data and zero/background correction is checked. There is something wrong here.

Lines 331-312: "…the experiment would only have underestimated $N_2$ production processes…" – surely these are two of the three processes you are investigating?!

*Figs/tables*

There seems to be some overlap/repetition between figures, please try and summarise data in fewer figures.

Keep colours consistent for N species between figures (i.e. $NO_3^-$ appears in red, blue, yellow)

Why is there already so much 15N-$N_2$ at the time when the tracer is added (e.g. Fig 2)? I suggest the data and zero/background correction is checked. There is something wrong here.

Are both fig 3 and 6 necessary as they are quite similar? Comparing them it seems like there is much more $N_2O$-denit and DNRA but hardly any $N_2$-denit in fig 6 than is shown in Fig 3's 15N recovery. Perhaps you can double check:

$^{29}N_2 = 1 \times {}^{15}N$,

$^{30}N_2 = 2 \times {}^{15}N$,

$^{45}N_2O = 1 \times {}^{15}N$

$^{46}N_2O = 2 \times {}^{15}N$

Is Table 1 necessary as it is just copied from Salk et al?

---

## Referee Comment (RC2) · Oksana Coban (Referee) · 20 Mar 2020

The manuscript of Valiente et al. investigates pathways of nitrate removal in a saline lake with a particular focus on exploring the role of DNRA and anammox. As these processes have not been yet investigated sufficiently in saline lakes, scientific novelty is substantial. In general, authors have performed quite comprehensive experiments and did a fairly good job on thoroughly reporting and discussing the results. However, the manuscript is lacking a good structure, especially in the Results and Discussion section. Also, Tables and Figures should be reconsidered. I also encourage authors to check the manuscript for use of abbreviations and definitions and consistency of the terms you use. I therefore recommend a major revision of this otherwise very interesting study.

*Major comments:*

Line 26: 'N$_2$O-denitrification' is unclear term. Is it incomplete denitrification to N$_2$O or the last step of denitrification (N$_2$O reduction)? Although you explain it later in the introduction, it should be clear in the abstract itself.

Lines 26-29: from these two sentences it looks like DNRA was important both under light and in dark. So it is not clear how you make the conclusion in your next sentence about coupling on the anammox and DNRA.

Line 47: removing fixed N by producing N$_2$ and N$_2$O gas.

Line 60-63: it is not clear as an advance of recent studies at what exactly (supposedly lower than 0.25 mg/L?) O$_2$ concentrations can *nosZ* function; you should provide the concentrations, otherwise this sentence does not make any sense.

Lines 75-78: these sentences are not build logically; a previous sentence does not support the following, and the role of light on coupled DNRA-anammox is not well explained

Line 90-91: your hypothesis is not very clear from the practical point of view. Would these results help to calculate mass balance of a saline lake? Or what is the ultimate goal for the measurements based on this assumption? Elaborate more clear research objectives.

Line 135: why were the mesocosms incubated at +25 °C? What conditions is this temperature representative of?

Line 148: what is the percentage (atom-%) of $^{15}NO_3^-$ in added $NO_3^-$?

Line 155: how much water was taken?

Line 163: why was salinity not measured?

Line 237: this chapter of the results and discussion consists of the results only. Furthermore it is not easy to follow when the information about differences between phases and treatments is given so early at the beginning. In this case a reader has to return constantly from the following subchapters of the discussion to this, first subchapter. I suggest that you incorporate this statistical information into the subchapters 3.2 and 3.3 when discussing results of a specific parameter.

You have too many figures and not all of them provide important enough information to be in the main body of the paper. I suggest you to move Figure 3 to Supplementary information. Also, rethink other ones.

Lines 239-240: you did not provide in the M&M how you measured salinity

Line 264: what do you mean by 'N$_2$-anammox' here? That suggests like there is another end product of anammox possible?

Line 314-315: you should explain what are the possible nitrogen converting processes that produce $^{45}N_2O$ and what processes result in $^{46}N_2O$. Furthermore, it doesn't look like $^{45}N_2O$ and $^{46}N_2O$ increased at all after time 15 hours. Then your assumption about $^{15}N$ recirculation by coupled DNRA nitrification does

not seem to be supported by the data. Instead you should find an explanation that would fit increase in $N_2O$ concentration but not in the $^{15}N$ in $N_2O$.

Line 324: the same comment with the previous one, you should assign specific processes to $^{29}N_2$ and $^{30}N_2$ production.

Lines 340-341: why would there be an increase in release of $CO_2$ and organic acids after your incubations as compared to natural conditions? Please explain.

Lines 369-371: I guess you could make a more robust assumption here about the $N_2O$ as a product of partial nitrification based on evidence that $N_2O$ concentration was increasing over the incubation time but not the $^{15}N$ in $N_2O$. I suggest you rethink this and probably also make calculations to assume quantitative contribution of other sources (such as nitrification) to $N_2O$ production.

Table 3: it is not clear what you mean here by 'canonical anammox' and '$N_2$-anammox'. You also do not explain this in the text.

*Minor comments:*

Lines 53-54: this sentence does not seem necessary.

Line 59: this sentence seem disconnected from the previous ones. Use a connector like 'also' or 'furthermore'

Line 84: it is questionable if a paper from 2003 can be called 'recent'

Line 85: it isn't clear here why anammox was underestimated. It is more logical to place this sentence at the end of line 88.

Lines 253-254: you should state here what kind of differences (i.e., where pH was found to be the highest, and where the lowest).

Lines 337-338: these changes were not statistically significant (Line 248-250), therefore this discussion does not seem necessary.

Line 345: it is not clear here to what ANOVA results you are referring to.

*Abbreviations (examples of misuse):*

Line 22, 90, 188, 191, 274: 'nitrate' should be abbreviated

Line 27: no need to introduce 'N' abbreviation here as you don't use it in the abstract anymore.

Line 50: dinitrogen should be $N_2$

Line 60, 86: nitrous oxide should be '$N_2O$'

Line 173, 400: replace 'nitrogen' with 'N'

---

## Referee Comment (RC3) · Anonymous Referee #3 · 2 Apr 2020

Review of Valiente et al. for Biogeosciences

The paper presents the relative contribution of several microbial processes related to nitrogen removal (denitrification, DNRA, and anammox) under different experimental conditions with/without light and with/without oxygen. The authors use 15N-isotope pairing technique to obtain the corresponding contribution. I consider that this paper is interesting, taking into account the global problem of nitrogen inputs in aquatic ecosystems and their consequences for $N_2O$ emissions. However, I found the paper wordy, hard to read, and the figures poorly illustrate the information about this work. I have several concerns that I detail below.

**Major concerns:**
-Quality of the figures

I think that the authors have to remake the figures to illustrate their results better. Figures 1, 2, and 4 include the acclimation phase that, in my opinion, masks the "real" results. I suggest showing in these figures only the exact experimental time (i.e., since the addition of the 15N-nitrate, time 0). Most changes occur in the first 24h. I think these figures could be more illustrative showing only these 24h-36h that is the period until NO3 disappears. The complete figures could go to supplementary material.

Figure 3 should be cut at the same time that Figures 1,2 and 4. In this figure, I am also wondering if the specific times with more than 100% of recovery are errors. Please, consider deleting these times. For instance, time 3 in the treatment OD that reach almost 120%. I think there is an error mostly because time 2.5 and time 4 are pretty similar.

-The precision of some chemical analysis.
The concentration of ammonium shows tremendous values of standard deviation (Figure 1). I have concerns to assume the low replicability of this analysis. The concentration of DOC and its changes is high. I have some concerns here too. I did not see the complete protocol for DOC. In saline, endorheic lakes DOC is usually high (around 1-10 mM), but the values around 40 mM are extremely high even for this type of systems. Water from saline lakes needs a protocol for DOC with longer purge time and acid additions to make sure all the DIC has been removed. I was unable to see these details in the methods of acid addition and purge time. Changes in 20 mM are so extreme that I have concerns. Usually, phytoplankton blooms can change almost nothing DOC concentration in water.

**Minor concerns**

- Lines 42-48, I think this paragraph is not necessary since the paper is about nitrates removing and confuse the reader.
- Lines 100-108, I found some incoherence here. Is Petrola lake submitted to nitrogen inputs or not?
- Line 158, What is dissolved bound nitrogen?

-Line 348, I think the reference of McCrackin and Elser is not about sediments

---

## Author Comment (AC1) · 22 Apr 2020

**Manuscript: bg-2020-20**

**Title:** Oxygen and light determine the pathways of nitrate reduction in sediments of a highly saline lake

**RESPONSE TO REFEREE #1**

We greatly appreciate the reviewer's comments, which have helped us to improve the manuscript. In general, we have restructured the manuscript and revised the figures and tables included, moving some of them to supplementary material. In addition, Judith Prommer was added in the co-author list due to her significant contribution in explaining the role of nitrification in the discussion section. Our responses are shown below the reviewer's comments in blue.

**Major general comments:**

As above, please re-consider the structure of how each section is approached and should be proofread. I also suggest it's less confusing to keep Results and discussion as separate sections.

We have restructured the manuscript following your recommendation: "Results and discussion" have been separated in "Results" (Section 3) and "Discussion" (Section 4).

Former point 3.1 is kept in "Results" section separated in two different sub-sections ("3.1 Differences between treatments in chemical parameters" and "3.3 Measured rates of N-loss processes"). In addition, the relevant information from former point 3.2 ("Hydrogeochemical dynamics during sediment incubations") has been moved to "Results" (as "3.2 Hydrochemical evolution"), whereas the remaining part is included in the first sub-section of "Discussion" ("4.1 N-removal over time").

Check nitrate/NO3 - nitrite/NO2 - throughout for consistency

Both ($NO_3^-$ and $NO_2^-$) followed a confusing and heterogeneous notation throughout the manuscript. We have reviewed and corrected it, as well as other chemical compounds that were not completely homogeneous in their notation (i.e. $NH_4^+$, $N_2O$, $N_2$).

Consider the relevance of references you use… some are from water column (e.g. Jensen et al 2011) or sediment with very difference settings.

Thank you for the comment. We acknowledge that Jensen's paper is on water column processes. Nevertheless, we decided to include this reference as it described a very particular interface (the ocean's oxygen minimum zone) where oxygen is limited, similar to the conditions found in the water-sediment interface.

About the references of sediment studies that we provided, we agree that they cover very different settings. However, the literature addressing the three N-removal processes that we studied (denitrification, DNRA and anammox) is very limited in (saline) lake ecosystems. Therefore, we were forced to include references from many different lacustrine and/or saline locations in order to compare our results.

Title: Add in 'sediment' somewhere

We added "sediment" in the title, which has been modified so that: Oxygen and light determine the pathways of nitrate reduction in sediments of a highly saline lake

**Introduction**

The introduction is quite disjointed with no logical direction to draw the reader in. For future potential submissions I would suggest a nicer structure to introduce the reader to your topic – e.g.:

- Shortly introduce N cycle and identify large anthropogenic impacts

- Overview on N cycling processes in sediments, focus on nitrate reducing processes/end products

- Introduce saline lakes and their importance/why are they interesting/understudied

- Potential factors controlling $NO_3^-$ reducing processes in this lake (light, O2 etc) (I should say that I didn't realize the study was only on sediments until line 80 as this in not explicitly mentioned before. Additionally, Jensen et al 2011 is a water column study so might not be relevant to sediments (where there should anyway be plenty of NH4+)).

Thank you so much for this valuable comment, not only for this manuscript, but also for future ones. Therefore, the introduction has been rewritten and restructured as:

- First paragraph about N cycle and $NO_3^-$ impacts

- Second and third paragraphs about $NO_3^-$ removal pathways in sediments (stating that the study is in sediments)

- Fourth paragraph about quantification of such processes

- Fifth paragraph about saline lakes

- Sixth paragraph explaining the goal of the study.

**Methods**

Please re-think the structure and need for the amount of text here

A lot of the text in the Methods section is unnecessary. If you are referencing a method from another paper, it should only be very briefly described in your methods (e.g. a lot of things don't need to be reiterated from Salk et al).

Thanks. We have removed text from the "Methods" section, especially in sub-sections "2.2 Sediment incubations" and "2.4 Isotope composition of N species". In addition, the Table showing IPT calculations following Salk et al. has been moved to Supplementary Information.

What is the relevance of the treatments to your study site? Does the lake become stratified/anoxic in some months? What is the phytobenthos? Diatoms? Bacterial mats? This needs to be put into context with better descriptions of your site (in methods and results)

We added such information to section 2.1 ("Study site") and included in the discussion ("4.1 N-removal over time") with the following texts:

**Section 2.1**: *The lake is shallow (maximum depth 2 m), with major water volume oscillations depending on seasonal precipitation, and is not stratified, with a regular mixing throughout the year caused by both wind and cooling. Phytoplankton is present in the water column of Pétrola Lake and includes diatoms (Amphora spp., Nitzschia spp.), cyanobacteria (Oscillatoria spp., Phormidium spp.), and green algae (Chlamydomonas spp., Tetraselmis spp.) (in Spanish: Confederación Hidrográfica del Segura, unpublished data)… Despite that the Pétrola endorheic basin was declared vulnerable to $NO_3^-$ pollution by the Regional Government of Castilla-La Mancha in 1998, it still receives a continuous supply of N mainly derived from inorganic synthetic fertilizers (Valiente et al., 2018). As a result, eutrophication of the water layer occurs leading to the dominance of phytoplankton, reducing light levels, and promoting bottom-water oxygen depletion because of bacterial decomposition.*

**Section 4.1**: *Three different treatments were applied during sediment incubations by modifying oxygen and light conditions in the water column. The darkness treatment mimics the reduction of light derived from enhanced development of planktonic organisms, as commonly observed in shallow eutrophic lakes (Cristofor et al., 1994). In shallow lakes, wind-driven water mixing contributes to avoid anaerobic bottom water (Utsumi et al., 1998). However, shallow eutrophic lakes may exhibit extreme fluctuations in DO concentrations, undergoing anoxia as a result of the collapse of phytoplankton blooms (Robarts et al., 2005) together with high sediment oxygen demand (Mallin et al., 2006). These conditions are found in Pétrola Lake, and therefore, the study of the treatments explained above in this study were: OL (oxygen + light), OD (oxygen + darkness), and AD (anoxia + darkness).*

Section 2.1: What depth were water samples taken?

Surface water samples were collected for the study, what has been added into Section 2.1 (line 115). In addition, a better description of the sampling point has been added (line 112): *The sampling site (control point 2651 in Valiente et al., 2018) was deep enough (approximately 50 cm) to allow us to sample sediment cores and overlying water, but located close to the lake's depocenter without any direct input of polluted freshwater streams or wastewaters. We therefore consider it representative of the natural conditions of the lake.*

Nutrient samples should be filtered (at least 0.2, possibly 0.45um filters) so no nutrients are produced/consumed between sampling/measuring.

To analyze in situ conditions, samples were filtered through 0.45 um. Therefore, such information has been included and the text is as follows (line 115): *To evaluate initial in situ natural conditions (NC), surface water samples were collected, filtered through 0.45 μm pore size nylon filters and stored at 4 °C in darkness prior to further analyses.*

Those samples to refill the mesocosms once arrived to the laboratory were not filtered.

Welti et al use a reservoir to feed the sealed (gas-space-free) mesocosms, I'm very confused about the method description here, was the mesocosm water itself bubbled?

Experiments were adapted from Welti's design. However, in our mesocosms we did not use a reservoir to feed them (we directly bubbled them). So, in line 121, the following sentence has been rewritten: *Mesocosm preparation for core incubations was adapted from previous works (Welti et al., 2012), except for the use of a feeding water reservoir.*

In addition, in Section 2.2 we have added the following sentence (line 129): *In the lab, each mesocosm was filled with lake water and bubbled with either air (oxic treatment) or argon (anoxic treatment).*

Section 2.2:

The first paragraph is a very long way to write 'the overlying water of each mesocosm was bubbled with either air (oxic treatment) or Argon (anoxic treatment)

Thanks. According to your recommendations, we have reduced the length of the paragraph and such information has been summarized in the sentence included above (line 129).

Why do you seal mescosms with no air space if you're going to bubble them anyway?

You're right. The reason to bubble them was to keep the anoxic conditions in the anoxic treatments (bubbled with Argon), but unnecessary in the oxic treatments. However, this sentence has been removed in order to summarize information from Section 2.2.

Line 135: what do you mean by pump? Do you mean a wheel/stirrer to mix the water to avoid stagnation? If not please explain more clearly – and add how the mesocosm water was mixed.

Yes, a small aquarium pump was used to mix the water and avoid stagnation, which has been included in the text (line 132): *a small aquarium pump was installed in the inner wall to prevent stagnation.*

Was light intensity monitored/measured?

Light intensity was not monitored. The reason for not doing so was that the room where the incubations were performed is not directly exposed to sunlight. Natural light entered through a window, in front of which the mesocosms of the light treatment (OL) were uniformly exposed. In addition, the days when the incubations were performed had clear skies (27.07.15 to 01.08.15), so we could assume that the intensity of light received must have been uniform over time.

It's fine to use Dalsgaard et al 2000 for timing calculations and assuming 1 mm oxygen penetration. However an oxygen penetration depth of 1 mm is not correct for the anoxic treatments. Why was this very high resolution time series chosen? More reasoning should be presented behind this as it seems a bit unnecessary. What is the relevance of 24 h anoxia and darkness in a 2 m deep lake? Diel and seasonal conditions should be better described in terms of biogeochemistry and phytobenthos etc.

The sampling frequency was adjusted on the basis of the oxic treatment, assuming 1 mm of oxygen penetration. Of course, this approach is not valid for an anoxic treatment, where there is no oxygen able to penetrate into the sediment. However, to facilitate comparison between treatments, we decided to adopt the same sampling frequency for all treatments.

We considered that a high resolution of sampling was necessary at the beginning of the experiment, to better know the fate of the tracer added. However, due to the complexity of the processes that can co-occur, finishing our experiment within the first 24 hours would have led us to miss valuable information about the processes that are triggered secondarily (as seen, for example, in the DNRA-anammox coupling). As a practical matter, we decided to decrease the sampling frequency after the first 24 hours.

With regard to the treatment of anoxia and darkness, as noted before, in order to compare the treatments on equal terms, it was necessary to maintain the same sampling frequency. In these shallow, non-stratified and eutrophic lakes, when algae blooms develop, light cannot penetrate (darkness treatment),

which together with a high rate of microbial decomposition of organic matter leads to anoxic conditions (anoxic treatment). These conditions of darkness and anoxia can be maintained over times beyond daily cycles.

Was water removed though sampling the mesocosms replaced? Were dilution effects accounted for?

The volume of water removed wasn't replaced in the mesocosms. The option of making an artificial matrix was discarded due to the chemical complexity of the matrix we were working with. Furthermore, we discarded the addition of filtered or otherwise preserved lake water, as it could lead to a change in the nature of the experiment. So, the final concentrations showed in this manuscript were corrected considering the effect of sample removal.

In the anoxic treatment why do you assume all NO3- reduction takes place in the sediment? Why not also the water column?

Thank you for that interesting comment. The role that microorganisms in the water column can play in the AD treatment was not properly explained. Therefore, the paragraph talking about diatom-bacteria aggregates (section 4.1, line 324) has been modified and a new reference has been included (Kamp et al., 2011), which explains how they are able to survive conditions of darkness and anoxia. This paragraph is written as follows:

*Based on isotope data, these processes seem to mainly account for the reduction of the added $NO_3^-$. However, traceability can sometimes be problematic in $^{15}N$-IPT studies due to processes such as uptake and intracellular storage (Robertson et al., 2019). Significant inputs of $NO_3^-$ may also promote blooms of diatoms (frequent in Pétrola Lake), which are physiologically adapted to grow rapidly under $NO_3^-$ rich conditions (Bronk et al., 2007). A phytoplankton bloom was observed after $^{15}NO_3^-$ addition in the light treatment (OL), with a subsequent decrease. Even though we cannot prove it, the role of diatom-bacteria aggregates should be considered. These are able to survive anoxia and darkness by reducing $NO_3^-$ from the water column to $NH_4^+$ (DNRA), subsequently fueling benthic anaerobic N-cycling (Kamp et al., 2011; 2016), which could alter the time pattern of $^{15}N$ in $NO_3^-$ and $NH_4^+$. Therefore, under anoxia and darkness, the reduction of $NO_3^-$ may not be limited to the sediments.*

Line 164-165: Is this the name of the site? This has not been introduced/mentioned until now.

Yes, 2651 refers to the control point where samples were collected. This sampling point was already described in a previous study (Valiente et al., 2018), and has been better described as stated previously in this document (line 112).

Section 2.4: A lot of this text is unnecessary and can be streamlined.

Following your recommendation, this section has been considerably shortened.

Line 181: Use original references for the microdiffusion method.

The reference to Brooks et al. (1989) has been included in Section 2.4 (line 183).

**Results and Discussion** (I suggest it's better to split into two sections)

This section is very confusing to read so please consider re-structuring. All of 3.1 seems to be results and 3.2 onwards is a mixture of results/discussion.

Following your recommendation, and as stated at the beginning of this document, "Results and Discussion" has been separated in "3. Results" and "4. Discussion". Former section 3.2 has been also divided, as part of it was mainly results description.

It's also important to include the in situ conditions you measured and a description of the site. What is the relevance of a longer anoxic incubation to your site?

The in situ conditions (or natural conditions in the text, referred as NC) are included in Table 1. As these samples were collected at time -48 h, they are notated as $NC_{-48}$. The relevance of long anoxic incubation in this site has been previously explained, as well as the conditions particular conditions of the sampling site.

What are the subscript numbers? E.g. OL72, t(2), F(2,6)

Thanks for the question. Those subscripts for NC, OL, OD and AD refer to the time of sampling. So, $OL_{72}$ means sampling in the OL treatment at time 72 h. This has also been added to the legend of Table 1.

Subscripts following $t$ indicate the degrees of freedom for the t-test (per definition, $n - 1$). In the case of ANOVA, the subscript numbers for F indicate the degrees of freedom between-sample ($h - 1$) and within-sample ($n - h$).

Don't forget units (e.g. lines 262-270)

Thanks for the reminder. The values presented in such lines (F-values) correspond to those obtained in the F-test of the ANOVA. These values are obtained by calculating the ratio between the explained variance (or between-group variability) and the unexplained variance (or within-group variability), both in the same units. Therefore, F-values have no units.

In general, a more thorough discussion of data is needed.

We also agree that our manuscript needed some more discussion, especially regarding the dynamics of $N_2O$ (section 4.2). We have made a detailed interpretation of the evolution of $^{45}N_2O$ and $^{46}N_2O$ (Figure 2) to try to explain alternative mechanisms of $N_2O$ production apart from denitrification. As a result, the following two paragraphs have been added (lines 368-394):

*Studies involving the role of $N_2O$-denitrification in saline aquatic environments are mainly restricted to marine ecosystems. Our high measured rates may be explained by the high biological activity after $^{15}NO_3^-$ addition, in the absence of nutrient limitation and/or low $N_2O$ reductase activity. Nonetheless, the different patterns observed for $^{29}N_2$ and $^{45}N_2O$ (Figure 2) cannot be explained, if denitrification was the sole source of $N_2$ and $N_2O$, in which case the proportions of $^{29}N_2$ and $^{30}N_2$ would match the proportions of $^{45}N_2O$ and $^{46}N_2O$ assuming steady state conditions (Trimmer et al., 2006). Differences in $^{29}N_2$ and $^{45}N_2O$ can be attributed to anammox, which can imbalance the proportion of $^{15}N$ by producing $^{29}N_2$. However, nitrification also produces $N_2O$ during its first step. This step involves the oxidation of ammonia ($NH_3$) to $NO_2^-$ by either ammonia-oxidizing archaea (AOA) or ammonia-oxidizing bacteria (AOB). AOB contain two distinct $N_2O$-producing pathways. The first mechanism, referred to as "hybrid formation" involves the combination of one N atom from $NO_2^-$ and one from $NH_4^+$ or an intermediate of its oxidative metabolism,*

*such as hydroxylamine ($NH_2OH$) or nitric oxide (NO) (Kozlowski et al., 2016; Frey et al., 2019). The other mechanism is the "nitrifier-denitrification" pathway that sequentially oxidizes $NH_4^+$ to $NO_2^-$, which is then reduced to NO and $N_2O$ (Wrage et al., 2001; Frame and Casciotti, 2010).*

*A possible explanation is the $^{15}N$ recirculation by coupled DNRA-nitrification (DNRA fueling nitrification to $N_2O$), which is a process whose importance has recently been highlighted in estuarine sediments (Dunn et al., 2009; Murphy et al., 2016). Although treatments OD and OL meet the conditions for this process to take place, this assumption is not fully supported by $^{45}N_2O$ and $^{46}N_2O$ evolution over time. $^{45}N_2O$ did show an increase over time, but not $^{46}N_2O$ (Figure 2). In addition, the vast majority of $N_2O$ measured during the incubation was $^{44}N_2O$, as the sum of $^{45}N_2O$ + $^{46}N_2O$ did not account for the huge $N_2O$ concentration at the end of the experiments (0.5 mmol/L in OL, above 2.0 mmol/L in OD and AD; Figure 1). DNRA would produce $^{15}NH_4^+$, which would be subsequently either oxidized-reduced by the nitrifier-denitrification mechanism or combined by the hybrid pathway with existing $^{15}NO_2^-$ (from our tracer addition). In both scenarios, it would result in $^{46}N_2O$ due to the merge of two $^{15}N$ atoms. Alternatively, there could be a direct coupling between externally supplied $^{15}NO_2^-$ and internally $^{14}N$ by ammonia-oxidizers. As stated in the previous section, $NH_4^+$ increased as a result of OM mineralization, supplying the $^{14}NH_4^+$ source. This $^{14}NH_4^+$, coupled to $^{15}NO_2^-$, can form $^{45}N_2O$ by the hybrid pathway as shown by previous studies (Trimmer et al., 2016). The large amount of $^{44}N_2O$ formed can be derived by $^{14}NH_4^+$ formed by OM mineralization and further processing by the nitrifier-denitrification mechanism, which is preferred under reduced oxygen conditions (Frame and Casciotti, 2010). To reveal the contribution of $N_2O$ production linked to ammonia oxidation by AOA and AOB, we tried to calculate gross nitrification.*

The 'stages' 1, 2 and 3 have not been defined/introduced until the results, please re-consider how you refer to the experiments.

Thank you for your comment. They have been included in section 2.2 (lines 144-153) when talking about the different timing of the incubations. In addition, we thought that the notation could be misleading, and we've changed by: initial stage or S0 (from sampling to tracer addition); middle stage or S1 (from tracer addition to 24 h); and final stage or S2 (from 24 h to the end of the incubation).

Why is there already so much 15N-N2 at the time when the tracer is added (in Fig 2)? I suggest the data and zero/background correction is checked. There is something wrong here.

According to your suggestion, we have checked the data we presented in Figure 2. For $N_2O$ ($^{45}N_2O$ and $^{46}N_2O$) it is clear that at time 0 the quantity of $N_2O$ is the lowest, and increased to reach more or less a stationary state (but not zero). The explanation for the oscillations are given in the section 4.2, as stated above, linked not only to denitrification but also to other hidden processes (like nitrification). For the $N_2$ the increase is not visible. Due to the rapid denitrification activity, equilibrium is reached quickly. In fact, concentrations of $^{29}N_2$ and $^{30}N_2$ are close to those of atmospheric air. In spite of that, the value measured at time 0 is not relevant for rate calculations because we worked using the slope value at each specific point for $N_2$ and $N_2O$, as explained in section 2.5 (line 208).

Lines 331-312: "…the experiment would only have underestimated N2 production processes…" – surely these are two of the three processes you are investigating?!

This sentence has been removed as it could lead to misunderstanding. The authors wanted to explain that the mass balance was close to 100% (explained in former Figure 3), and any deviation from that could lead to little underestimation in the total $N_2$ production.

**Figs/tables**

There seems to be some overlap/repetition between figures, please try and summarize data in fewer figures.

Thanks for the suggestion. Following your recommendations, we have moved former figures 3 (mass balance) and 4 (physico-chemical evolution) to Supplementary Information.

Keep colors consistent for N species between figures (i.e. NO3- appears in red, blue, yellow)

Thank you for informing us of this detail. We have solved this inconsistency and kept the same colors for each N species (following those used in Figure 2).

Why is there already so much 15N-N2 at the time when the tracer is added (e.g. Fig 2)? I suggest the data and zero/background correction is checked. There is something wrong here.

We have already made an explanation for this observation in the previous section of the revised document.

Are both fig 3 and 6 necessary as they are quite similar? Comparing them it seems like there is much more N2O-denit and DNRA but hardly any N2-denit in fig 6 than is shown in Fig 3's 15N recovery. Perhaps you can double check:

29N2 = 1 x 15N,

30N2 = 2 x 15N,

45N2O = 1 x 15N

46N2O = 2 x 15N

Figure 3 has transferred to Supplementary Information. However, Figure 3 and Figure 6 explain different things. On the one hand, Figure 3 represents the mass balance by the recovery of $^{15}N$ from our measurements. $^{15}N$ was added as $NO_3^-$ but was measured in 4 different chemical compounds ($NO_3^-$, $NH_4^+$, $N_2O$, $N_2$). Any measurement of concentration and $^{15}N$ enrichment comes with an error. The accumulation of steps can inflate the total error on the final values. Therefore, showing values at each time point so close to 100% actually demonstrates how precisely all the measurements were performed.

On the other hand, Figure 6 represents the proportion of each $NO_3^-$ reduction pathway ($N_2$-denitrification, $N_2O$-denitrification, DNRA, and anammox) to the total contribution at each time. This calculation is made on the basis of the total $NO_3^-$ reduction.

Is Table 1 necessary as it is just copied from Salk et al?

Table 1 is adapted from Salk et al. (2017), but not just copied from Salk. DNRA parameters included in our table are obtained from the equations provided in the Supplementary Information of Salk's work. So, you are right about the necessity of including it in the main manuscript and we decided to move it to the Supplementary Information (Table S1).

**References**

Brooks, P.D., Stark, J.M., McInteer, B.B. and Preston, T.: Diffusion Method To Prepare Soil Extracts For Automated Nitrogen-15 Analysis, Soil Science Society of America Journal, 53, 1707-1711, doi:10.2136/sssaj1989.03615995005300060016x, 1989.

Cristofor, S., Vadineanu, A., Ignat, G. and Ciubuc, C.; Factors affecting light penetration in shallow lakes, Hydrobiologia, 275(1), 493-498, doi:10.1007/BF00026737, 1994.

Frame, C.H. and Casciotti, K.L.: Biogeochemical controls and isotopic signatures of nitrous oxide production by a marine ammonia-oxidizing bacterium, Biogeosciences, 7, 2695-2709, doi:10.5194/bg-7-2695-2010, 2010.

Frey, C., Bange, H.W., Achterberg, E.P., Jayakumar, A., Löscher, C.R., Arévalo-Martínez, D.L., León-Palmero, E., Sun, M., Xie, R.C., Oleynik, S. and Ward, B.: Regulation of nitrous oxide production in low oxygen waters off the coast of Peru. Biogeosciences Discussions, 1-35, doi:10.5194/bg-2019-476, in review, 2019.

Jensen, M. M., Lam, P., Revsbech, N. P., Nagel, B., Gaye, B., Jetten, M. S. and Kuypers, M. M.: Intensive nitrogen loss over the Omani Shelf due to anammox coupled with dissimilatory nitrite reduction to ammonium, The ISME Journal, 5(10), 1660–1670, doi:10.1038/ismej.2011.44, 2011.

Kamp, A., de Beer, D., Nitsch, J.L., Lavik, G. and Stief, P.: Diatoms respire nitrate to survive dark and anoxic conditions, Proceedings of the National Academy of Sciences, 108 (14), 5649-5654, doi:10.1073/pnas.1015744108, 2011.

Kozlowski, J.A., Stieglmeier, M., Schleper, C., Klotz, M.G. and Stein, L.Y.: Pathways and key intermediates required for obligate aerobic ammonia-dependent chemolithotrophy in bacteria and Thaumarchaeota, The ISME journal, 10(8), 1836-1845, doi:10.1038/ismej.2016.2, 2016.

Mallin, M. A., Johnson, V. L., Ensign, S. H. and MacPherson, T. A.: Factors contributing to hypoxia in rivers, lakes, and streams, Limnology and Oceanography, 51(1), doi:10.4319/lo.2006.51.1_part_2.0690, 2006.

Salk, K. R., Erler, D. V., Eyre, B. D., Carlson-Perret, N. and Ostrom, N. E.: Unexpectedly high degree of anammox and DNRA in seagrass sediments: Description and application of a revised isotope pairing technique, Geochimica et Cosmochimica Acta, 211, 64–78, doi:10.1016/j.gca.2017.05.012, 2017.

Trimmer, M., Chronopoulou, P.M., Maanoja, S.T., Upstill-Goddard, R.C., Kitidis, V. and Purdy, K.J.: Nitrous oxide as a function of oxygen and archaeal gene abundance in the North Pacific, Nature communications, 7(1), 1-10, doi:10.1038/ncomms13451, 2016

Utsumi, M., Nojiri, Y., Nakamura, T., Nozawa, T., Otsuki, A. and Seki, H.: Oxidation of dissolved methane in a eutrophic, shallow lake: Lake Kasumigaura, Japan, Limnology and Oceanography, 3, doi:10.4319/lo.1998.43.3.0471, 1998.

Valiente, N., Carrey, R., Otero, N., Soler, A., Sanz, D., Muñoz-Martín, A., Jirsa, F., Wanek, W. and Gómez-Alday, J. J.: A multi-isotopic approach to investigate the influence of land use on nitrate removal in a highly saline lake-aquifer system, Science of The Total Environment, 631–632, 649–659, doi:10.1016/j.scitotenv.2018.03.059, 2018.

Welti, N., Bondar-Kunze, E., Mair, M., Bonin, P., Wanek, W., Pinay, G. and Hein, T.: Mimicking floodplain reconnection and disconnection using [15]N mesocosm incubations, Biogeosciences, 9(11), 4263–4278, doi:10.5194/bg-9-4263-2012, 2012.

Wrage, N., Velthof, G.L., van Beusichem, M.L. and Oenema, O.: Role of nitrifier denitrification in the production of nitrous oxide, Soil Biology and Biochemistry, 33 (12-13), 1723-1732, doi:10.1016/S0038-0717(01)00096-7, 2001.

---

## Author Comment (AC2) · 22 Apr 2020

**Manuscript: bg-2020-20**

**Title:** Oxygen and light determine the pathways of nitrate reduction in sediments of a highly saline lake

**RESPONSE TO REFEREE #2 (Oksana Coban)**

We would like to thank the reviewer for her valuable comments, which we believe have helped us a lot to improve the manuscript in general, and also some aspects of the discussion. In general, we have restructured the manuscript, rewritten some parts and revised the figures and tables included, moving some of them to supplementary material. In addition, Judith Prommer was added in the co-author list due to her significant contribution in explaining the role of nitrification in the discussion section. Our responses are shown below the reviewer's comments in blue.

**Major comments:**

Line 26: 'N2O-denitrification' is unclear term. Is it incomplete denitrification to N2O or the last step of denitrification (N2O reduction)? Although you explain it later in the introduction, it should be clear in the abstract itself.

Thank you for the comment. This term is now explained in the abstract by changing "$N_2O$-denitrification" to "denitrification to $N_2O$", what we consider adequately explains which step of denitrification is involved without unnecessarily lengthening the abstract. The term $N_2O$-denitrification is then first mentioned in the introduction and later used for "partial" denitrification.

Lines 26-29: from these two sentences it looks like DNRA was important both under light and in dark. So it is not clear how you make the conclusion in your next sentence about coupling on the anammox and DNRA.

Thanks, we agree that it needed a clarification. So, we have changed to "*DNRA, and especially denitrification to $N_2O$, were the dominant nitrogen (N) removal pathways when oxygen and/or light were present (up to 82%). In contrast, anoxia and darkness promoted $NO_3^-$ reduction by DNRA (52%) combined to N loss by anammox (28%).*"

Line 47: removing fixed N by producing N2 and N2O gas.

Thanks, of course. We have included $N_2O$ in the description.

Line 60-63: it is not clear as an advance of recent studies at what exactly (supposedly lower than 0.25 mg/L?) O2 concentrations can *nosZ* function; you should provide the concentrations, otherwise this sentence does not make any sense.

We were not able to provide the concentrations such authors used, as the concentrations are not indicated (Wittorf et al., 2016). Therefore, we decided to include the genera of bacteria able to do it by the following sentence (line 45):

*… recent studies showed the presence of nosZ gene or nosZ transcripts in potentially non-denitrifying genomes of aerobic genera like Gemmatimonas (Orellana et al., 2014; Yoon et al., 2016; Hallin et al., 2018)…*

Lines 75-78: these sentences are not build logically; a previous sentence does not support the following, and the role of light on coupled DNRA-anammox is not well explained

Thank you, we agree with your comment. The "Introduction" section has been reorganized and some changes have been made, including putting coupled DNRA-anammox into context of the different $NO_3^-$ removal pathways described in aquatic sediments.

Line 90-91: your hypothesis is not very clear from the practical point of view. Would these results help to calculate mass balance of a saline lake? Or what is the ultimate goal for the measurements based on this assumption? Elaborate more clear research objectives.

During the reorganization of the introduction, the last paragraph has been rewritten explaining the rationale of the present study (line 85):

As described above, oxygen plays a key role in favoring certain processes over others. In addition, light availability can impact the balance between $NO_3^-$ removal pathways as light will enhance primary production and the production of dissolved oxygen. Here, we tested the hypothesis that oxic and light exposed conditions in the water column promote denitrification over DNRA and anammox. For this purpose, we incubated lacustrine sediments from a eutrophic saline lake (Pétrola Lake, Spain) and applied the revised $^{15}N$-IPT to confirm and quantify N-cycling rates. Taken together, these findings not only improve our knowledge of the mass balance of N pollutants in saline lakes, but also of how they contribute to climate change in terms of $N_2O$ release.

Line 135: why were the mesocosms incubated at +25 oC? What conditions is this temperature representative of?

The reason to choose 25ºC was that this value was the mean water temperature of samples collected in Pétrola Lake in the summer months of the 2013 campaign, as shown in Valiente et al. (2018). By 2015, this was the last field campaign with comprehensive summer data, and so we relied on that data, which has been cited in the manuscript (line 134).

Line 148: what is the percentage (atom-%) of 15NO3 in added NO3?

$^{15}N$-labeled $NO_3^-$ was 98 atom% at $^{15}N$. It has been added to the manuscript (line 148).

Line 155: how much water was taken?

We collected 20 mL for inorganic N concentrations and N isotope compositions, and 10 mL for physico-chemical, DOC and DNb. This information has also been added (lines 154-156).

Line 163: why was salinity not measured?

We used Total Dissolved Solids (TDS) as estimator for salinity (Williams, 1966), although we are aware that the use of total dissolved salts is preferable to determine salinity in this type of waters (Boerlage, 2012). However, this determination required equipment which the laboratory where incubations were performed did not possess. Consequently, a sentence has been included in the manuscript to indicate that salinity was estimated from TDS (line 163).

Line 237: this chapter of the results and discussion consists of the results only. Furthermore it is not easy to follow when the information about differences between phases and treatments is given so early at the beginning. In this case a reader has to return constantly from the following subchapters of the discussion to this, first subchapter. I suggest that you incorporate this statistical information into the subchapters 3.2 and 3.3 when discussing results of a specific parameter.

We have restructured the manuscript following your recommendation: "Results and discussion" have been separated in "Results" (Section 3, including all the statistical information) and "Discussion" (Section 4). Former point 3.1 is kept in "Results" section separated in two different sub-sections ("3.1 Differences between treatments in chemical parameters" and "3.3 Measured rates of N-loss processes"). In addition, the relevant information from former point 3.2 ("Hydrogeochemical dynamics during sediment incubations") has been moved to "Results" (as "3.2 Hydrochemical evolution"), whereas the remaining part is included in the first sub-section of "Discussion" ("4.1 N-removal over time").

You have too many figures and not all of them provide important enough information to be in the main body of the paper. I suggest you to move Figure 3 to Supplementary information. Also, rethink other ones.

Thanks for the suggestion. Following your recommendations, we have moved former figures 3 (mass balance) and 4 (physico-chemical evolution) to Supplementary Information.

Lines 239-240: you did not provide in the M&M how you measured salinity

As answered above, the use of TDS for estimate salinity has been added to the "Materials and Methods" section.

Line 264: what do you mean by 'N2-anammox' here? That suggests like there is another end product of anammox possible?

This term has been explained and the sentence clarified (line 288). The reason to introduce this term was to make a difference between $N_2$ produced by denitrification and $N_2$ produced by anammox. From this point, the latter will be referred as $N_2$-anammox.

Line 314-315: you should explain what are the possible nitrogen converting processes that produce 45N2O and what processes result in 46N2O. Furthermore, it doesn't look like 45N2O and 46N2O increased at all after time 15 hours. Then your assumption about 15N recirculation by coupled DNRA nitrification does not seem to be supported by the data. Instead you should find an explanation that would fit increase in N2O concentration but not in the 15N in N2O.

Thank you so much for this comment, it gave us the opportunity to better explain the mechanisms occurring during our incubations. As you said, our assumption of $^{15}N$ recirculation by coupled DNRA and nitrification is not well supported by the data. It could lead to $^{46}N_2O$ (and $^{30}N_2$, discussed below). However, there was a slight increase in $^{45}N_2O$, not observed in $^{46}N_2O$, and especially in $^{44}N_2O$ when accounting for the total $N_2O$ concentration (Figure 1). This imbalance cannot be explained if denitrification was the sole source of both $N_2O$ and $N_2$.

$^{46}N_2O$ can be produced by denitrification and, as mentioned above, by coupled DNRA-nitrification. Produced $^{15}NH_4^+$ by DNRA may be subsequently nitrified, either by the nitrifier-denitrification mechanism or combined by the hybrid pathway with existing $^{15}NO_2^-$. In both scenarios, it would result in $^{46}N_2O$ due to the merge of two $^{15}N$ atoms. This trend is not observed in our experiments.

Concerning $^{45}N_2O$, its production could be linked to a direct coupling between externally supplied $^{15}NO_2^-$ (reduced $^{15}NO_3^-$) and internally converted $^{14}N$ by ammonia-oxidizers. It was noticed that $NH_4^+$ increased over time in the mesocosms as a result of OM mineralization. This mineralization would supply the $^{14}NH_4^+$ source, which can be combined with $^{15}NO_2^-$ to form $^{45}N_2O$ by the hybrid pathway, as shown by previous studies (Trimmer et al., 2016). The large amount of $N_2O$ at the end of the experiment must be $^{44}N_2O$. It could be formed by $^{14}NH_4^+$ formed by OM mineralization and further processing by the nitrifier-denitrification mechanism, which is preferred under reduced oxygen conditions.

These explanations have been included in the manuscript (lines 368-394) with the following text:

*Studies involving the role of $N_2O$-denitrification in saline aquatic environments are mainly restricted to marine ecosystems. Our high measured rates may be explained by the high biological activity after $^{15}NO_3^-$ addition, in the absence of nutrient limitation and/or low $N_2O$ reductase activity. Nonetheless, the different patterns observed for $^{29}N_2$ and $^{45}N_2O$ (Figure 2) cannot be explained, if denitrification was the sole source of $N_2$ and $N_2O$, in which case the proportions of $^{29}N_2$ and $^{30}N_2$ would match the proportions of $^{45}N_2O$ and $^{46}N_2O$ assuming steady state conditions (Trimmer et al., 2006). Differences in $^{29}N_2$ and $^{45}N_2O$ can be attributed to anammox, which can imbalance the proportion of $^{15}N$ by producing $^{29}N_2$. However, nitrification also produces $N_2O$ during its first step. This step involves the oxidation of ammonia ($NH_3$) to $NO_2^-$ by either ammonia-oxidizing archaea (AOA) or ammonia-oxidizing bacteria (AOB). AOB contain two distinct $N_2O$-producing pathways. The first mechanism, referred to as "hybrid formation" involves the combination of one N atom from $NO_2^-$ and one from $NH_4^+$ or an intermediate of its oxidative metabolism, such as hydroxylamine ($NH_2OH$) or nitric oxide (NO) (Kozlowski et al., 2016; Frey et al., 2019). The other mechanism is the "nitrifier-denitrification" pathway that sequentially oxidizes $NH_4^+$ to $NO_2^-$, which is then reduced to NO and $N_2O$ (Wrage et al., 2001; Frame and Casciotti, 2010).*

*A possible explanation is the $^{15}N$ recirculation by coupled DNRA-nitrification (DNRA fueling nitrification to $N_2O$), which is a process whose importance has recently been highlighted in estuarine sediments (Dunn et al., 2009; Murphy et al., 2016). Although treatments OD and OL meet the conditions for this process to take place, this assumption is not fully supported by $^{45}N_2O$ and $^{46}N_2O$ evolution over time. $^{45}N_2O$ did show an increase over time, but not $^{46}N_2O$ (Figure 2). In addition, the vast majority of $N_2O$ measured during the incubation was $^{44}N_2O$, as the sum of $^{45}N_2O$ + $^{46}N_2O$ did not account for the huge $N_2O$ concentration at the end of the experiments (0.5 mmol/L in OL, above 2.0 mmol/L in OD and AD; Figure 1). DNRA would produce $^{15}NH_4^+$, which would be subsequently either oxidized-reduced by the nitrifier-denitrification mechanism or combined by the hybrid pathway with existing $^{15}NO_2^-$ (from our tracer addition). In both scenarios, it would result in $^{46}N_2O$ due to the merge of two $^{15}N$ atoms. Alternatively, there could be a direct coupling between externally supplied $^{15}NO_2^-$ and internally $^{14}N$ by ammonia-oxidizers. As stated in the previous section, $NH_4^+$ increased as a result of OM mineralization, supplying the $^{14}NH_4^+$ source. This $^{14}NH_4^+$, coupled to $^{15}NO_2^-$, can form $^{45}N_2O$ by the hybrid pathway as shown by previous studies (Trimmer et al., 2016). The large amount of $^{44}N_2O$ formed can be derived by $^{14}NH_4^+$ formed by OM mineralization and further processing by the nitrifier-denitrification mechanism, which is preferred under reduced oxygen conditions (Frame and Casciotti, 2010). To reveal the contribution of $N_2O$ production linked to ammonia oxidation by AOA and AOB, we tried to calculate gross nitrification.*

Line 324: the same comment with the previous one, you should assign specific processes to 29N2 and 30N2 production.

The explanation for this question is similar to that described for $^{45}N_2O$ and $^{46}N_2O$. An explanation of the processes leading to $^{30}N_2$ and $^{29}N_2$ production has been added into section 4.1 (line 342).

The production of $^{30}N_2$ can be attributed either to denitrification of $^{15}NO_3^-$, or to coupled DNRA-anammox, by combining the DNRA substrate ($^{15}NO_2^-$) with the DNRA product ($^{15}NH_4^+$) (Holtappels et al., 2011). About $^{29}N_2$, both anammox and denitrification can contribute (Song et al., 2016). For the first one (canonical anammox), existing $^{15}NO_2^-$ can be combined with present $^{14}NH_4^+$, which can be produced by OM mineralization. However, denitrification can play an important role if this formed $^{14}NH_4^+$ by OM mineralization is subsequently nitrified.

Lines 340-341: why would there be an increase in release of CO2 and organic acids after your incubations as compared to natural conditions? Please explain.

Microbial decomposition produces organic acids and $CO_2$ from the breakdown of larger organic carbon molecules (e.g. Herndon et al., 2015). As a result of the addition of $NO_3^-$ as electron acceptor, and considering that enough organic matter is available to donate electrons, an increase in the microbial metabolism is expected, and as a result, higher release of $CO_2$ than in natural conditions. The decrease in pH (Table 1) at the end of the incubations we understand was in line with this assumption.

Lines 369-371: I guess you could make a more robust assumption here about the N2O as a product of partial nitrification based on evidence that N2O concentration was increasing over the incubation time but not the 15N in N2O. I suggest you rethink this and probably also make calculations to assume quantitative contribution of other sources (such as nitrification) to N2O production.

Thank you so much for your suggestion. As discussed above, we consider that nitrification had a relative influence on the production of $N_2O$. For this reason, we calculated gross nitrification and gross $NO_3^-$ consumption rates based on isotope pool dilution (IPD) theory, using $^{15}N$ at% of $NO_3^-$ on 10 time intervals per mesocosm. Once these rates were calculated, we wanted to cross our values with published values of $N_2O$ production rates by ammonia oxidizers belonging to AOA and AOB, to derive a maximum estimate for nitrifier $N_2O$ emissions and contributions to overall $N_2O$ production. However, gross nitrification rates were below detection limit, therefore obviating the possibility to estimate nitrifier $N_2O$. For this reason, we consider that unfortunately nitrification was undetectable in this type of mesocosm experiment, probably because it was produced using $^{14}NH_4^+$ as argued above. To summarize our calculations and assumptions, we added (lines 394-395):

*Unfortunately, the obtained rates were below LOD, meaning that another type of mesocosm experiments would be needed to measure the contribution of ammonia oxidizers to $N_2O$ production (which was not the main focus of this study).*

Table 3: it is not clear what you mean here by 'canonical anammox' and 'N2-anammox'. You also do not explain this in the text.

The term "canonical anammox", used in Salk et al. (2017), refers to the anammox process which consumes non-DNRA-derived $NH_4^+$. This explanation has been included in the "Material and Methods" section (line 212).

**Minor comments:**

Lines 53-54: this sentence does not seem necessary.

Thanks, this sentence has been removed.

Line 59: this sentence seem disconnected from the previous ones. Use a connector like 'also' or 'furthermore'

Thank you for your comment. As listed above, the Introduction sections has been reorganized and the first of these sentences removed from this paragraph.

Line 84: it is questionable if a paper from 2003 can be called 'recent'

"Recently" has been removed from this sentence.

Line 85: it isn't clear here why anammox was underestimated. It is more logical to place this sentence at the end of line 88.

Thanks, this sentence has been moved to the end of the paragraph as you suggested.

Lines 253-254: you should state here what kind of differences (i.e., where pH was found to be the highest, and where the lowest).

After the values of the ANOVA analysis, the following sentence has been added (lines 240-241):

*At the end of the experiment, the highest mean pH values were found in the oxic treatments, significantly higher than mean pH measured in AD treatment (Table 1).*

Lines 337-338: these changes were not statistically significant (Line 248-250), therefore this discussion does not seem necessary.

We agree with your suggestion and part of the sentence has been removed. However, the changes we talk about are related to the temporal evolution of DOC from stage S1 (sharp increase after tracer addition) to the end of the experiment. Therefore, although there were no significant differences between treatments at the end of the experiments, we considered necessary to keep part of the discussion as follows (line 334):

*A sharp increase of DOC, probably derived from a bloom collapse, was observed in all the treatments during S1 stage (Figure S3). Afterwards, DOC concentration decreased as a result of heterotrophic metabolism. DON values also support this, as the decreasing percentages of DON:DNb underline the role of OM remineralization throughout the incubation.*

Line 345: it is not clear here to what ANOVA results you are referring to.

A reference to section 3.3 has been added. In this section, the second paragraph describes differences in N-processes within each treatment, and the co-dominant role of $N_2O$-denitrification is shown.

**Abbreviations (examples of misuse):**

Line 22, 90, 188, 191, 274: 'nitrate' should be abbreviated

Line 27: no need to introduce 'N' abbreviation here as you don't use it in the abstract anymore.

Line 50: dinitrogen should be N2

Line 60, 86: nitrous oxide should be 'N2O'

Line 173, 400: replace 'nitrogen' with 'N'

Thank you very much for indicating us those examples. These abbreviations followed in general a confusing and heterogeneous notation throughout the manuscript. So, we have reviewed and corrected them, including all the chemical compounds that were not completely homogeneous in their notation (i.e. $NH_4^+$, $N_2O$, $N_2$).

**References**

Boerlage, S.F.: Measuring salinity and TDS of seawater and brine for process and environmental monitoring—which one, when?, Desalination and Water Treatment, 42(1-3), 222-230, doi: 10.1080/19443994.2012.683191, 2012.

Herndon, E.M., Mann, B.F., RoyChowdhury, T., Yang, Z., Wullschleger, S.D., Graham, D., Liang, L. and Gu, B.: Pathways of anaerobic organic matter decomposition in tundra soils from Barrow, Alaska, Journal of Geophysical Research: Biogeosciences, 120, 2345– 2359, doi:10.1002/2015JG003147, 2015.

Holtappels, M., Lavik, G., Jensen, M. M. and Kuypers, M. M. M.: Chapter ten - 15N-Labeling Experiments to Dissect the Contributions of Heterotrophic Denitrification and Anammox to Nitrogen Removal in the OMZ Waters of the Ocean, in Methods in Enzymology, vol. 486, edited by M. G. Klotz, pp. 223–251, Academic Press., 2011.

Orellana, L. H., Rodriguez-R, L. M., Higgins, S., Chee-Sanford, J. C., Sanford, R. A., Ritalahti, K. M., Löffler, F. E., and Konstantinidis, K. T.: Detecting nitrous oxide reductase (nosZ) genes in soil metagenomes: method development and implications for the nitrogen cycle, MBio, 5(3), e01193-14, doi: 10.1128/mBio.01193-14, 2014.

Salk, K. R., Erler, D. V., Eyre, B. D., Carlson-Perret, N. and Ostrom, N. E.: Unexpectedly high degree of anammox and DNRA in seagrass sediments: Description and application of a revised isotope pairing technique, Geochimica et Cosmochimica Acta, 211, 64–78, doi:10.1016/j.gca.2017.05.012, 2017.

Song, G. D., Liu, S. M., Kuypers, M. M. M. and Lavik, G.: Application of the isotope pairing technique in sediments where anammox, denitrification, and dissimilatory nitrate reduction to ammonium coexist, Limnology and Oceanography: Methods, 14(12), 801–815, doi:10.1002/lom3.10127, 2016.

Trimmer, M., Chronopoulou, P.M., Maanoja, S.T., Upstill-Goddard, R.C., Kitidis, V. and Purdy, K.J.: Nitrous oxide as a function of oxygen and archaeal gene abundance in the North Pacific, Nature communications, 7(1), 1-10, doi: 10.1038/ncomms13451, 2016

Valiente, N., Carrey, R., Otero, N., Soler, A., Sanz, D., Muñoz-Martín, A., Jirsa, F., Wanek, W. and Gómez-Alday, J. J.: A multi-isotopic approach to investigate the influence of land use on nitrate removal in a highly saline lake-aquifer system, Science of The Total Environment, 631–632, 649–659, doi:10.1016/j.scitotenv.2018.03.059, 2018.

Williams, W.D.: Conductivity and the concentration of total dissolved soilds in Australian lakes, Marine and Freshwater Research, 17(2), 169-176, 1966.

Wittorf, L., Bonilla-Rosso, G., Jones, C.M., Bäckman, O., Hulth, S. and Hallin, S.: Habitat partitioning of marine benthic denitrifier communities in response to oxygen availability, Environmental Microbiology Reports, 8, 486-492, doi:10.1111/1758-2229.12393, 2016.

---

## Author Comment (AC3) · 22 Apr 2020

**Manuscript: bg-2020-20**

**Title:** Oxygen and light determine the pathways of nitrate reduction in sediments of a highly saline lake

**RESPONSE TO REFEREE #3**

We are grateful with the referee's comments, which were very helpful to improve the manuscript. In general, we have restructured the manuscript, rewritten some parts and revised the figures and tables included, moving some of them to supplementary material. In addition, Judith Prommer was added in the co-author list due to her significant contribution in explaining the role of nitrification in the discussion section. Our responses are shown below the reviewer's comments in blue.

**Major concerns:**

-Quality of the figures

I think that the authors have to remake the figures to illustrate their results better. Figures 1, 2, and 4 include the acclimation phase that, in my opinion, masks the "real" results. I suggest showing in these figures only the exact experimental time (i.e., since the addition of the 15N-nitrate, time 0). Most changes occur in the first 24h. I think these figures could be more illustrative showing only these 24h-36h that is the period until NO3 disappears. The complete figures could go to supplementary material.

We agree that results could be illustrated better. For this reason, we have deleted the stabilization phase from the main figures. Following your recommendations, Figure 1 now starts with time 0 (from the time of tracer addition), as well as the remaining figures of the main manuscript. The complete time incubation of N-species and the evolution of physico-chemical parameters (former Figure 4) have been moved to Supplementary Information (Figures S1 and S3, respectively).

About the end time of the figures included in the main manuscript, we decided to keep the whole incubation time. The reason for doing that is the valuable information we could miss by showing only this 24-36 h period. It's the case of the $N_2O$ production in treatments OD and AD at the end of the experiments (Figure 1). Such concentrations (above 2 mmol/L) are presumably produced not only by denitrification but also by partial nitrification, fueled by OM mineralization (section 4.2, lines 368-395). This is a process triggered secondarily that, in our view, is important to discuss in the manuscript (especially as a source of $N_2O$) and hence our decision to keep the plots until 72 h of incubation.

Figure 3 should be cut at the same time that Figures 1,2 and 4. In this figure, I am also wondering if the specific times with more than 100% of recovery are errors. Please, consider deleting these times. For instance, time 3 in the treatment OD that reach almost 120%. I think there is an error mostly because time 2.5 and time 4 are pretty similar.

Figure 3 has been moved to Supplementary Information (Figure S3). About the information represented here, it refers to the mass balance (the recovery of $^{15}N$) from our measurements. $^{15}N$ was added as $NO_3^-$ but was measured in 4 different chemical compounds ($NO_3^-$, $NH_4^+$, $N_2O$,

N$_2$). Any measurement of concentration and [15]N enrichment comes with an error, implying a loss of [15]N (in gray) or an excess of it (as the case of time 3 in treatment OD). The accumulation of steps can inflate the total error on the final values. Therefore, the general recovery showing values at each time point so close to 100% actually demonstrates how precisely all the measurements were performed.

-The precision of some chemical analysis.

The concentration of ammonium shows tremendous values of standard deviation (Figure 1). I have concerns to assume the low replicability of this analysis. The concentration of DOC and its changes is high. I have some concerns here too. I did not see the complete protocol for DOC. In saline, endorheic lakes DOC is usually high (around 1-10 mM), but the values around 40 mM are extremely high even for this type of systems. Water from saline lakes needs a protocol for DOC with longer purge time and acid additions to make sure all the DIC has been removed. I was unable to see these details in the methods of acid addition and purge time. Changes in 20 mM are so extreme that I have concerns. Usually, phytoplankton blooms can change almost nothing DOC concentration in water.

Thank you for the comment. About DOC analysis, we followed the methodology provided by the manufacturer in the equipment manual, which is also briefly discussed in Stubbins and Dittmar (2012). So, samples were acidified to pH ≈ 2 with 2 M HCl. The optimal purging time for samples to remove all DIC after acidification was determined by measuring a series of identical samples in triplicates until results for DOC were achieved within a 95% confidence interval. The necessary time was determined to be 5 minutes, therefore this was the purging time used for all samples, even the ones with lower dissolved solids content. A sentence including this information has been included in the methodology.

**Minor concerns**

- Lines 42-48, I think this paragraph is not necessary since the paper is about nitrates removing and confuse the reader.

We agree with that, and therefore, these lines have been removed from the introduction.

- Lines 100-108, I found some incoherence here. Is Petrola lake submitted to nitrogen inputs or not?

Thank you for the comment, it is certainly something we did not express correctly. The last part of that paragraph has been rewritten (lines 105-111) as follows:

*The lake has been classified as a heavily modified water body due to the inputs of agricultural pollutants as well as untreated wastewater directly spilled from Pétrola Village. Despite that the Pétrola endorheic basin was declared vulnerable to NO$_3^-$ pollution by the Regional Government of Castilla-La Mancha in 1998, it still receives a continuous supply of N mainly derived from inorganic synthetic fertilizers (Valiente et al., 2018). As a result, eutrophication of the water layer occurs leading to the dominance of phytoplankton, keeping out the light, and promoting bottom-water oxygen depletion because of bacterial decomposition.*

- Line 158, What is dissolved bound nitrogen?

Dissolved bound nitrogen (DNb) refers to the sum of dissolved N species (NO$_3^-$, NO$_2^-$, NH$_4^+$, organic N) excluding gaseous N forms (e.g. N$_2$ and N$_2$O). A short note has been added to Section 2.3 (line 175).

- Line 348, I think the reference of McCrackin and Elser is not about sediments

We checked this reference, where the authors "measured the rate of denitrification and nitrous oxide ($N_2O$) production during denitrification in sediments from 32 Norwegian lakes at the high and low ends of a gradient of atmospheric N deposition". Therefore, we have decided to keep the reference in the manuscript.

**References**

McCrackin, M. L. and Elser, J. J.: Atmospheric nitrogen deposition influences denitrification and nitrous oxide production in lakes, Ecology, 91(2), 528–539, doi:10.1890/08-2210.1, 2010.

Stubbins, A. and Dittmar, T.: Low volume quantification of dissolved organic carbon and dissolved nitrogen, Limnology and Oceanography Methods, 10, doi:10.4319/lom.2012.10.347, 2012.

Valiente, N., Carrey, R., Otero, N., Soler, A., Sanz, D., Muñoz-Martín, A., Jirsa, F., Wanek, W. and Gómez-Alday, J. J.: A multi-isotopic approach to investigate the influence of land use on nitrate removal in a highly saline lake-aquifer system, Science of The Total Environment, 631–632, 649–659, doi:10.1016/j.scitotenv.2018.03.059, 2018.